# Loss of adaptive capacity in asthmatic patients revealed by biomarker fluctuation dynamics after rhinovirus challenge

Anirban Sinha[1,2,3]*, René Lutter[1,2], Binbin Xu[4], Tamara Dekker[2], Barbara Dierdorp[2], Peter J Sterk[1], Urs Frey[3], Edgar Delgado Eckert[3]

[1]Department of Respiratory Medicine, Amsterdam UMC, University of Amsterdam, Amsterdam, Netherlands; [2]Department of Experimental Immunology, Amsterdam UMC, University of Amsterdam, Amsterdam, Netherlands; [3]Department of Biomedical Engineering and University Children's Hospital, University of Basel, Basel, Switzerland; [4]University of Bordeaux, Inserm, Bordeaux Population Health Research Center, UMR 1219, Bordeaux, France

**Abstract** Asthma is a dynamic disease, in which lung mechanical and inflammatory processes interact in a complex manner, often resulting in exaggerated physiological, in particular, inflammatory responses to exogenous triggers. We hypothesize that this may be explained by respiratory disease-related systems instability and loss of adaptability to changing environmental conditions, manifested in highly fluctuating biomarkers and symptoms. Using time series of inflammatory (eosinophils, neutrophils, FeNO), clinical and lung function biomarkers (PEF, FVC, $FEV_1$), we estimated this loss of adaptive capacity (AC) during an experimental rhinovirus infection in 24 healthy and asthmatic human volunteers. Loss of AC was estimated by comparing similarities between pre- and post-challenge time series. Unlike healthy participants, the asthmatic's post-viral-challenge state resembled more other rhinovirus-infected asthmatics than their own pre-viral-challenge state (hypergeometric-test: p=0.029). This reveals loss of AC and supports the concept that in asthma, biological processes underlying inflammatory and physiological responses are unstable, contributing to loss of control.

*For correspondence:
a.sinha@amsterdamumc.nl

## Introduction

The quantitative study of physiologic systems, such as the respiratory system, has revealed their ability to maintain a highly organized internal environment that is fluctuating within certain limits, despite being constantly exposed to a variable external environment (*Que et al., 2001*; *Tirone and Brunicardi, 2001*; *Palmer and Clegg, 2016*). The term homeokinesis has been coined to describe this ability, substituting the concept of homeostasis (*Goldstein and Kopin, 2017*) to emphasize that fluctuations in the internal environment are normal (*Macklem, 2008*; *Macklem and Seely, 2010*; *Yates, 1982*; *Eke et al., 2002*; *Glass, 2001*). Homeokinesis is at the core of the observed adaptability, that is, adaptive capacity, of physiologic systems in response to changing environmental conditions (*Goldberger et al., 2002*). It is part of the remarkable complexity characteristic of such systems, which is believed to originate from non-linear interactions and feedback-loops between their constitutive parts (*Goldberger et al., 2002*; *Garfinkel, 1983*; *Goldberger, 1996*; *Que, 1998*).

Over the past decades a considerable research effort has been invested into mathematically analyzing the fluctuation behavior of physiologic time series with the aim of characterizing the normal homeokinetic variability of physiologic systems (*Garfinkel, 1983*; *Glass and Mackey, 1988*;

*Goldberger et al., 2000*; *Glass and Kaplan, 1993*; *Amigó and Small, 2017*). One of the most counterintuitive findings has been that both excessive and too little variation are indicative of pathological modifications and aging (*Que et al., 2001*; *Goldberger et al., 2002* and references therein). Within this paradigm, a chronic disease, such as asthma, may be understood as changes in the system that render it either too rigid or overly unstable (*Frey et al., 2011*). Consequently, such disease or aging related changes of the system are accompanied by a loss of adaptive capacity (*Goldberger et al., 2002*). However, whether this can be detected in asthmatics using longitudinal measurements, that is time series, of asthma-related biomarkers has never been investigated.

The aim of this study is to test whether in asthmatics the adaptive capacity to a standardized environmental perturbation, such as an experimental viral challenge, is altered in comparison to healthy subjects.

In this prospective, longitudinally designed study comprising healthy and asthmatic subjects, we measured time series of a set of standard lung functional and inflammatory/immune biomarkers two months prior to and one month following an experimental rhinovirus 16 (RV16) infection induced by controlled and deliberate inoculation of healthy and asthmatic volunteers. This choice was driven by the fact that rhinovirus (RV) infections in asthmatics have been found to be among the most prominent external triggers of acute worsening of asthma symptoms, asthma exacerbations, and of loss of control (*Johnston et al., 1995*; *Nicholson et al., 1993*).

Quantifying a loss of adaptive capacity, that is, an impairment in the ability to cope with external perturbations, in a physiologic system will depend on how adaptive capacity is defined. This, in turn, depends on the physiological context. For instance, researchers have directly linked the capacity of rats to adapt to environmental heat stresses to the ability of the animal's liver cells to rapidly express the heat shock protein HSP70 in high quantities (*Hall et al., 2000*). Other scientists have suggested 'the capacity of a physiological system to bring itself autonomously back to the normal homeostatic range after a challenge' as a more workable definition of adaptive capacity (*Orešic and Vidal-Puig, 2014*). While still very general, the latter definition seems more suitable in a (patho-) physiological context in which the specific molecular mechanisms behind the adaptation processes cannot be easily laid out. Indeed, in geriatric medicine this definition has been widely used, and the term homeostenosis was introduced to describe aging related loss of adaptive capacity (*Taffett, 2003*; *Fossion et al., 2018*; *Yashin et al., 2007*; *Fried et al., 2009*).

Based on these ideas, we define adaptive capacity as the ability of a physiological system to autonomously return to the normal homeokinetic range after an external challenge. We implemented this definition quantitatively in our study by comparing the participants' pre- and post-viral-challenge time series of measurements of the aforementioned biomarkers. We hypothesized that, for a given biomarker, the post-viral-challenge time series of a study participant with an unimpaired adaptive capacity would resemble the same subject's pre-viral-challenge time series. Conversely, the post-viral-challenge time series of a study participant with an impaired adaptive capacity would be relatively distinct from the same subject's pre-viral-challenge time series. In order to test this hypothesis, hierarchical clustering was used to group time series according to their relative similarity in an unassuming, data-driven manner.

With this approach we found experimental evidence for the loss of adaptive capacity of the human respiratory system due to asthma.

## Results

### Experimental rhinovirus challenge while monitoring cohort participants

In all cohort participants (12 healthy and 12 asthmatic volunteers), the biomarkers/parameters listed in *Table 1* below were measured during two months before, and during one month immediately after deliberate experimental inoculation with rhinovirus, resulting in pre- and post-viral-challenge time series of each biomarker/parameter. Plots of the time series of each biomarker can be found in *Supplementary file 2*. For the healthy and the asthmatics groups separately, summary statistics of the average before the viral challenge (average over 2 months) and after the viral challenge (average over 1 month) of each of these biomarkers/parameters can be found in the Appendix.

**Table 1.** Biomarkers/parameters measured in each cohort participant during two months before, and during one month immediately after deliberate experimental inoculation with rhinovirus.

The corresponding sampling frequencies can be found in columns 2 and 3. See the Materials and methods section below for more details on the study design, and on the measurement procedures and laboratory assays used. FEV1: forced expiratory volume in one second. FVC: forced vital capacity. PEF: peak expiratory flow. FeNO: fractional expired concentration of nitric oxide.

| Biomarker or parameter | Sampling frequency before rhinovirus challenge | Sampling frequency after rhinovirus challenge |
|---|---|---|
| Lung function (FEV1, FVC, FEV1/FVC, PEF) | 2x daily | 2x daily |
| Exhaled Nitric Oxide (FeNO) | 3x weekly | 3x weekly |
| Eosinophil and neutrophil cell density in nasal lavage fluid | 1x weekly | 3x weekly |

## Hierarchical clustering of biomarker time series

In order to quantitatively establish the degree of similarity or 'proximity' between two time series of a given biomarker, we used the Earth Mover's Distance (EMD), which regards each of the time series as a univariate empirical distribution of the biomarker at hand (see Materials and methods and Appendix for more details). The pre- and post-challenge time series (also referred to as uninfected participant and infected participant, respectively) of individual biomarker time series from all participants (both healthy and asthmatics) were clustered using the EMD as the distance metric between the time series. The outcomes for the levels of exhaled nitric oxide (FeNO), and the percentage of eosinophils in nasal lavage fluid are presented here, whereas the results for the other biomarkers are presented in the Appendix.

### Time series of exhaled Nitric Oxide (FeNO)

Findings are summarized in *Table 2*. The corresponding dendrogram is depicted in *Figure 1*. In brief, we found three clusters. Cluster 1 consists of four time series stemming from two asthmatics.

**Table 2.** Composition, enrichment analysis, and grouping characteristics of the clusters found by comparison of each participant's pre- and post-challenge time series of FeNO.

Enrichment is marked in bold letters, depletion in italics; the corresponding p-values were calculated using the hypergeometric test. The empirical p-values for the proportion of pre- and post-pairs were calculated using simulated permutations (see Materials and methods section). A participant is fully represented in a given cluster if both their pre- and post-challenge time series of measurements are contained in the cluster. For example, the healthy participant 'P08H' is fully represented in Cluster 2, as both their pre- and post-challenge time series of FeNO measurements are members of Cluster 2 (see *Figure 1* below). Partial representation corresponds to the scenario in which only one of the two time series (pre- and post-challenge) is a member of the cluster. For instance, the asthmatic participant " P07A' is only partially represented in Cluster 2, because their pre-challenge time series of FeNO measurements is part of Cluster 2, whereas their post-challenge time series of FeNO belongs to Cluster 3 (see *Figure 1* below). See also the Materials and methods section for the definition of neighbors.

| Characteristic \ cluster number | Cluster 1 | Cluster 2 | Cluster 3 |
|---|---|---|---|
| Size (%) | 4 (8.33 %) | 26 (54.17 %) | 18 (37.5 %) |
| Fully represented healthy participants | 0 | 11 | 0 |
| Partially represented healthy participants | 0 | 1 | 1 |
| Fully represented asthmatic participants | 2 | 1 | 8 |
| Partially represented asthmatic participants | 0 | 1 | 1 |
| Number of time series from healthy participants (%) | 0 (0%) | 23 (88.46 %) | 1 (5.56 %) |
| p-value of enrichment/depletion in time series from healthy participants | 0.055 | 1.78E-09 | *1.15E-06* |
| Number of neighboring pre- and post-pairs (%) | 2 (100 %) | 4 (28.57 %) | 2 (20 %) |
| Empirical p-value (probability of observing, under the null hypothesis, the number of neighboring pre- and post-pairs found in the data, as listed in the previous row above) | 0.003 | 0.007 | 0.097 |

As can be read off of the dendrogram in *Figure 1* below, and of the distance matrix depicted in Panel C of Figure 2 (see Materials and methods section below), these two participants are prominently different from the rest (regarding their FeNO time series), and might be regarded as outliers. Cluster 2 contains more healthy participants than expected by chance. In other words, Cluster 2 is *enriched* in healthy participants. Conversely, due to the balanced design of the cohort (equal numbers of healthy and of asthmatic participants), Cluster 2 is also *depleted* of asthmatic participants, that is it contains fewer asthmatic participants than expected by chance. And finally, Cluster 3, which is enriched in asthmatic participants. While all, but one, of the time series from healthy participants are grouped together in Cluster 2, the vast majority of time series from asthmatic participants are split into two different Clusters, namely Clusters 1 and 3. This suggest a higher heterogeneity among the asthmatics. In Cluster 2, the tendency for infected participants to be clustered together with their corresponding uninfected counterpart is statistically significant (p-value=0.007, see *Table 2* below). This is not the case for Cluster 3. The difference in this regard between Cluster 2 (mainly healthy participants) and Cluster 3 (mainly asthmatic participants) is further underpinned by the fact that, on average, the cophenetic distances (see Materials and methods section for the definition of cophenetic distance) between the infected cluster members and their uninfected counterparts are statistically significantly lower in Cluster 2 when compared to Cluster 3 (p-value=0.033, one-tailed Mann-Whitney-U-test, see *Appendix 1—figure 3*).

The sub-clusters found within Clusters 2 and 3, respectively (marked with orange and blue rectangles in *Figure 1*), were analyzed in terms of enrichment in or depletion of pre- and post-challenge time series. The results are presented in *Table 3*. This analysis provides evidence for a statistically significant separation of pre- and post-challenge time series within Cluster 3. Indeed, the union of subclusters 3.1 and 3.2 is enriched in pre-challenge time series (p-value=0.029, see *Table 3* below), whereas subcluster 3.3 is enriched in post-challenge time series (p-value=0.029, see *Table 3* below). Such a separation cannot be observed within Cluster 2.

A bootstrap based sensitivity analysis of these findings can be found in the Appendix.

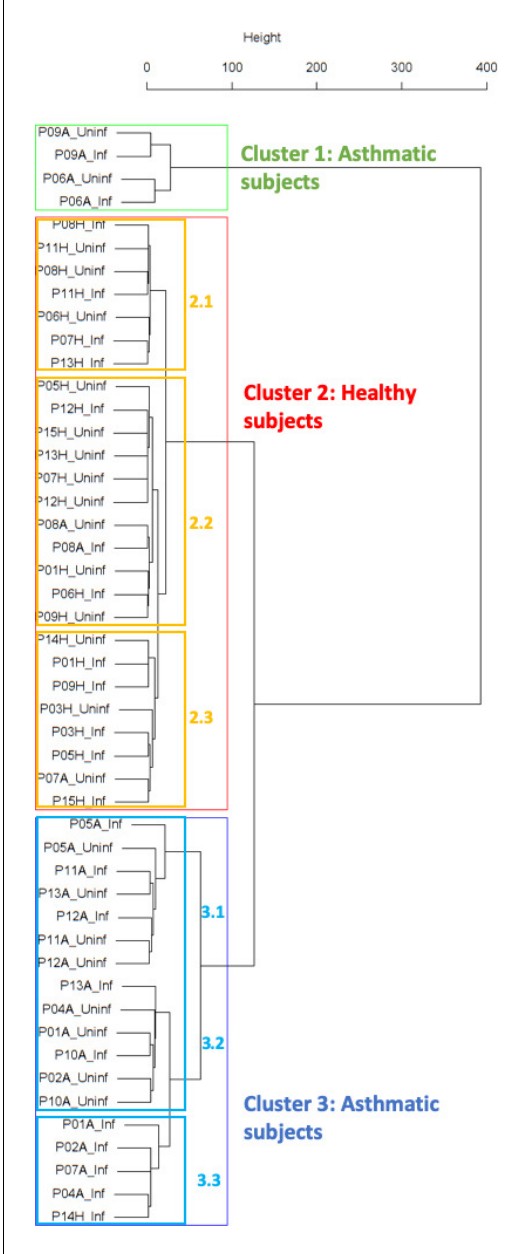

**Figure 1.** Cluster dendrogram obtained via hierarchical clustering of the participants' pre- and post-challenge time series of FeNO. The distance between any two-time series was calculated using the EMD. Rectangles mark the clusters and sub-clusters identified. From top to bottom: Cluster 1, Cluster 2 (subdivided into Clusters 2.1, 2.2, and 2.3), and Cluster 3 (subdivided into Clusters 3.1 and 3.2, and 3.3). Patient IDs are indicated by Pxy, their health status using H/A, denoting Healthy or Asthmatic, and their RV infection status by Uninf/Inf, which stands for Uninfected/Infected. Cluster 1 consists of time series from asthmatics which are prominently different from those from other asthmatic subjects in Cluster three and also from healthy subjects in Cluster 2. These might be regarded as outliers.

## Time series of percentage of eosinophils in nasal lavage fluid

Findings are summarized in *Table 4*. The corresponding dendrogram is depicted in *Appendix 1—figure 1*. In brief, three clusters were identified. Cluster 1 consists of four time series stemming from three asthmatics. As can be read off of the dendrogram depicted in *Appendix 1—figure 1*, these time series are prominently different from all the other time series, and might be regarded as outliers. Cluster 2 is enriched in healthy participants. And finally, Cluster 3, which is enriched in asthmatic participants. As seen in the analysis of FeNO, while the vast majority of the time series from healthy participants are grouped together in Cluster 2, most of the time series from asthmatic participants are split into two different Clusters, namely Clusters 1 and 3. This suggest a higher heterogeneity among the asthmatics. However, Cluster 1 in the eosinophil analysis and Cluster 1 in the FeNO analysis only have one asthmatic patient in common. Again, in Cluster 2, the tendency for infected participants to be clustered together with their corresponding uninfected counterpart is statistically significant (p-value=0.001, see *Table 4* below). This is not the case for Clusters 1 and 3. The difference in this regard between Cluster 2 (mainly healthy participants) and Cluster 3 (mainly asthmatic participants) is further substantiated by the fact that, on average, the cophenetic distances between the infected cluster members and their uninfected counterparts are statistically significantly lower in Cluster 2 when compared to Cluster 3 (p-value=8.96e-05, one-tailed Mann-Whitney-U-test, see *Appendix 1—figure 2*).

## Autocorrelation properties of the biomarker time series

For every participant, the autocorrelation coefficient of the lung function parameters time series and of the FeNO time series was calculated using a for each parameter type physiologically meaningful time lag. More specifically, a one-day lag was used for lung function parameters, and a two-days lag for FeNO. Due to the low sampling frequency used before the viral challenge, the time series of eosinophil and neutrophil cell density in nasal lavage fluid were not included in the autocorrelation analysis.

The resulting autocorrelation coefficients were then used to compare the groups of asthmatic and healthy participants prior to and after the viral challenge (see *Appendix 1—figures 10–14*). Briefly, in terms of autocorrelation, the lung function parameters PEF (% predicted), FVC, and FEV1/FVC discriminate significantly between the pre- and post-infection status in both healthy and asthmatic participants. Indeed, for these three parameters, there is a moderate positive autocorrelation before the challenge, which then disappears after the challenge. Furthermore, in terms of autocorrelation, only FeNO discriminates significantly between healthy and asthmatic participants, and it does so only after the challenge. More specifically, only asthmatics exhibit a moderate autocorrelation of their FeNO time series after the challenge. However, after a multiple pairwise-comparison correction aimed at controlling the false discovery rate, only the parameters FVC and FEV1/FVC discriminate significantly between the pre- and post-infection status in both healthy and asthmatic participants, while the other findings lose their statistical significance. The results after multiple pairwise-comparison correction are summarized in *Appendix 1—table 2*.

## Individual response to the viral challenge with respect to the biomarkers measured

In order to test the effectiveness of the virus challenge, we measured the individual patient's response with respect to each of the biomarkers measured. However, prior to doing that, the efficacy of the inoculation with RV16 needed to be established. Indeed, the results of blood antibody tests (RV16 seroconversion) along with RV Polymerase Chain Reaction (PCR) conducted on nasal lavage fluid taken from every participant after the inoculation indicated that 11 out of 12 healthy participants and 12 out of 12 asthmatics were effectively infected with the RV16 after inoculation (*Appendix 1—table 1*). According to the above mentioned laboratory tests, one healthy participant did not become infected. However, this participant did develop cold symptoms within a few days after the virus inoculation, suggesting that the laboratory tests failed to detect the ongoing infection although the participant was positively infected. Consequently, this participant was included in the analyses.

We then explored, for each of the biomarkers measured (listed in the first column of *Table 5*), for how many participants a statistically significant within-subject change upon infection can be

**Table 3.** Enrichment analysis of the sub-clusters found within the clusters described in **Table 2** above (the clusters marked with orange and blue rectangles in **Figure 1**).

Enrichment in pre-challenge time series is marked in bold letters, depletion of pre-challenge time series (and consequently enrichment in post-challenge time series) in italics; the corresponding p-values were calculated using the hypergeometric test.

| Sub-cluster number \ characteristic | Size (%) | Number of pre-challenge series (%) | p-value |
|---|---|---|---|
| Cluster 2.1 | 7 (26.92 %) | 3 (42.86 %) | 0.404 |
| Cluster 2.2 | 11 (42.31 %) | 8 (72.73 %) | 0.104 |
| Cluster 2.3 | 8 (30.77 %) | 3 (37.5 %) | 0.246 |
| Cluster 3.1 and 3.2 | 13 (72.22 %) | 8 (61.54 %) | **0.029** |
| Cluster 3.3 | 5 (27.78 %) | 0 (0.00 %) | *0.029* |

observed ('responders', see *Table 5*). To this end, two criteria for 'responders' were implemented. The first criterion, which regards time series as univariate empirical distributions of the biomarker at hand, aimed at detecting distributional changes in a given biomarker induced by the viral challenge: Here, each participant's pre- and post-challenge time series of each biomarker were compared using the Kolmogorov-Smirnov test. The second criterion aimed at detecting short-term and transient relative changes induced by the viral challenge in the context of the relative changes observed prior to the challenge. Here, throughout the entire period of observation, we assessed the relative change of each biomarker taking place within time intervals of 10 days. (see Subsection 5.2 and Figure 3 in the Materials and methods section below).

## Discussion

In this proof of concept study, we provided experimental evidence for the loss of adaptive capacity in the human respiratory system due to asthma. To this end, we hypothesized that a loss of adaptive capacity could be experimentally demonstrated by detection of a similarity diminution between the pre- and post-perturbation dynamics of the system. Using a data-driven clustering approach, we have shown that, in particular, FeNO and eosinophil time series were similar prior to and following the challenge in healthy subjects, suggesting stable homeokinetic behavior. In asthmatics, however, this similarity was predominantly reduced, suggesting a marked impact of the asthmatic condition on dynamic properties of the respiratory system, consistent with more unstable behavior and loss of adaptive capacity following the perturbation with viral infection. This loss of self-similarity is not merely the result of a larger response to the virus infection. Rather, we detected changes in the overall biomarker fluctuation dynamics elicited by the viral challenge that render asthmatics more similar to other infected asthmatic participants than to their uninfected counterparts.

### Experimental evidence supporting our hypothesis of a loss of adaptive capacity of the respiratory system in asthma

Our hypothesis in this study is based on the following question: For which type of participant, healthy or asthmatic, and for which biomarker is the disruption introduced by the viral challenge strong enough to render infected individuals more similar among themselves than to their uninfected counterparts? Our cluster analysis of the pre- and post-challenge time series of the percentage of eosinophils in nasal lavage fluid resulted in two main clusters: Cluster 2, which is statistically significantly enriched in healthy participants, and Cluster 3, which is mainly composed of asthmatic participants. In Cluster 2, the tendency for infected participants to be clustered together with their corresponding uninfected counterpart is statistically significant and clearly higher than in Cluster 3. In the clustering of the pre- and post-challenge time series of FeNO we found Cluster 2, mainly composed of healthy participants, and Cluster 3, made of nearly 95% asthmatics. Furthermore, within Cluster 3 we found a statistically significant separation of pre- and post-challenge time series. No such separation was found within Cluster 2. For both biomarkers (percentage of eosinophils in nasal lavage fluid and FeNO), the difference in this regard between Cluster 2 (mainly healthy participants)

**Table 4.** Composition, enrichment analysis, and grouping characteristics of the clusters found by comparison of each participant's pre- and post-challenge time series of percentage of eosinophils in nasal lavage fluid.

Enrichment is marked in bold letters, depletion in italics; the corresponding p-values were calculated using the hypergeometric test. The empirical p-values for the proportion of pre- and post-pairs were calculated using simulated permutations (see Materials and methods section). A participant is fully represented in a given cluster if both their pre- and post-challenge time series of measurements are contained in the cluster. Partial representation corresponds to the scenario in which only one of the two time series (pre- and post-challenge) is a member of the cluster. See also the Materials and methods section for the definition of neighbors.

| Characteristic \ cluster number | Cluster 1 | Cluster 2 | Cluster 3 |
|---|---|---|---|
| Size (%) | 4 (8.33 %) | 26 (54.17 %) | 18 (37.50 %) |
| Fully represented healthy participants | 0 | 11 | 1 |
| Partially represented healthy participants | 0 | 0 | 0 |
| Fully represented asthmatic participants | 1 | 2 | 7 |
| Partially represented asthmatic participants | 2 | 0 | 2 |
| Number of time series from healthy participants (%) | 0 (0%) | 22 (84.62 %) | 2 (11.11 %) |
| p-value of enrichment/depletion in time series from healthy participants | *0.055* | **1.09E-07** | *2.89E-05* |
| Number of neighboring pre- and post-pairs (%) | 1 (33.33 %) | 5 (38.46 %) | 1 (10 %) |
| Empirical p-value (probability of observing, under the null hypothesis, the number of neighboring pre- and post-pairs found in the data, as listed in the previous row above) | 0.123 | 0.001 | 0.424 |

and Cluster 3 (mainly asthmatic participants) is further substantiated by the fact that, on average, the cophenetic distances between the infected cluster members and their uninfected counterparts are statistically significantly lower in Cluster 2 when compared to Cluster 3.

In our clustering based on time series of cell density in nasal lavage fluid, there is a statistically significant separation of pre- and post-challenge time series. Moreover, within the subgroup of mainly post-challenge time series we found a cluster of size seven enriched in asthmatic participants (see *Appendix 1—figure 4*).

Summarizing, we have found evidence for the tendency of infected asthmatic participants to be more similar to other infected asthmatic participants than to their own uninfected counterparts when similarity is measured in terms of the biomarker dynamics of FeNO, of the percentage of eosinophils in nasal lavage fluid, and of cell density in nasal lavage fluid. This tendency was, however, not observed, when similarity was measured in terms of the biomarker dynamics of lung function parameters, and of neutrophil cell density in nasal lavage fluid.

## Physiological interpretation of the group differences in autocorrelation properties of the biomarker time series

We and others have previously shown that time series of lung function over days display a weak intrinsic autocorrelation over long- and short-time scales in asthmatic and healthy subjects (*Frey et al., 2005*; *Delgado-Eckert et al., 2018*; *Thamrin et al., 2016*). Lung function is correlated with lung function values of previous days; these correlation properties are related to severity of asthma and disease control (*Thamrin et al., 2016*). It has been hypothesized that such correlation properties are related to the balance between disease stability and adaptability of the system (*Frey et al., 2011*). We and others have previously shown that external stimuli such as medication can alter these correlation properties dependent on the applied drug action (*Frey et al., 2005*; *Thamrin et al., 2016*). Here, we provide first evidence that viral stimuli can also alter these correlation properties. In both healthy and asthmatic subjects, correlation of daily lung function was weakened by the viral challenge of the respiratory system. We hypothesize that following the viral challenge, the lung mechanical system properties are less deterministic consistent with a lower stability of the respiratory system.

**Table 5.** Proportions of responders within the groups of healthy and asthmatic participants, respectively.

Two different criteria were used in order to establish a statistically significant response. According to the first criterion, a participant is considered a responder with respect to a given biomarker if the outcome of comparing the pre-challenge time series and the post-challenge time series of the same biomarker by means of the Kolmogorov-Smirnov test results in a p-value<=0.05 (columns 2 and 3). According to the second criterion, a participant is considered a responder with respect to a given biomarker if the outcome of comparing, by means of a Mann-Whitney-U-test, the magnitude of relative changes observed during 10 day time intervals prior to the challenge with the magnitude of relative changes that took place during 10 day time intervals that contained the day of the challenge results in a p-value<=0.05 (columns 4 and 5). For calculating the proportion of responders within each group the p-values were corrected for multiple testing using the false discovery rate (FDR) method of Benjamini and Hochberg. FEV1: forced expiratory volume in one second. FVC: forced vital capacity. PEF: peak expiratory flow. FeNO: fractional expired concentration of nitric oxide. The lung function parameters FEV1 and FVC, and thereby their ratio FEV1/FVC, were normalized using the standardized reference equations recommended by Global Lung Function Initiative (GLI) Task Force for comparisons across different populations.

| Biomarker name | % Healthy responders (distributional changes) | % Asthmatic responders (distributional changes) | % Healthy responders (relative change within 10 days) | % Asthmatic responders (relative change within 10 days) |
|---|---|---|---|---|
| PEF (% of predicted) | 50.0% | 75.0% | 0.0% | 0.0% |
| Normalized FEV1 | 75.0% | 66.7% | 0.0% | 0.0% |
| Normalized FVC | 83.3% | 100.0% | 0.0% | 0.0% |
| Normalized FEV1/FVC | 75.0% | 66.7% | 16.7% | 0.0% |
| FeNO | 8.3% | 0.0% | 41.7% | 8.3% |
| Cell density in nasal lavage fluid | 0.0% | 0.0% | 66.7% | 41.7% |
| Neutrophils in nasal lavage fluid (%) | 0.0% | 0.0% | 25.0% | 16.7% |
| Eosinophils in nasal lavage fluid (%) | 8.3% | 0.0% | 0.0% | 16.7% |

## Asthma as a chronic disease: Causal chain of mechanisms or complex system behavior?

The classical analytical approach in asthma research is the identification of individual mechanisms (e. g., airway obstruction) or a series of mechanisms involved in the disease process (e.g. viral trigger → inflammation → bronchial hyperreactivity→ airway obstruction→ respiratory symptoms). However, epidemiological observations have questioned such a simple causal relationship between these mechanisms. For example, our previous work demonstrates that the strength of the trigger is often not proportional to the degree of response and symptoms (*Frey and Suki, 2008*). Also, the degree of inflammation, airway obstruction and bronchial responsiveness are often not closely related. Response to triggers (e.g., environmental pollutants) can occur with time lags and also in various degrees of intensity depending on the pre-existing conditions of the respiratory system (e.g. increased bronchial hyperresponsiveness following allergic sensitization or viral infection). Such a behavior is better explained by complex system behavior of a chronic disease. However, so far, the latter is difficult to capture and remained a theoretical concept.

We believe that the current experiment provided significant evidence to support the existence of such system behavior in asthma. In a well-established human challenge model, the respiratory system was challenged with a standardized (viral) stimulus, an established approach to test the behavior of complex network systems. We made the following observations: First, prior and after the challenge a set of lung functional and inflammatory asthma-biomarkers showed temporal fluctuations, in both healthy and asthmatic human subjects. There was a large inter and intra-individual variation. Second, the temporal relationship between inflammatory and lung functional biomarkers and symptoms was weak, not supporting the concept of simple proportional interactions of the above-mentioned causal chain of mechanisms. Third, despite these fluctuations and despite the absence of a strong response to the viral challenge in a distinct mechanistic biomarker, we identified differences between healthy and asthmatic humans in these dynamic variations. This indicates that the complex

interactions of inflammatory and lung functional parameters and thus the control of biological responses relevant to the respiratory system must be different in asthma. Our data support the hypothesis of a loss of adaptive capacity in asthma, which impedes the fast return to the pre-challenge stable dynamic steady state. We hypothesize that some clinical phenomena are consistent with such loss of adaptive capacity, such as, for example increased morbidity and prolonged respiratory symptoms after viral infection in asthmatics, persistent bronchial hyperresponsiveness after viral challenge, or slower return of airway obstruction following viral challenge (*Busse et al., 2010*).

## Variable and heterogeneous effect of the viral challenge on lung function and inflammatory/immune biomarkers

We carried out a quantitative characterization of individual response to the viral perturbation. This was done using two computational/statistical approaches. One approach aimed to capture the changes elicited by the viral challenge taking place over longer time periods (comparison of the pre- and post-challenge time series, viewed as empirical distributions), whereas the other assessed relative short-term changes occurring at shorter time scales (comparison of the magnitude of relative changes observed during 10 day time intervals).

There is a clear macroscopic/functional manifestation of the kindling RV infection, as reflected at the level of distributional changes induced by the viral challenge on lung function parameters. Indeed, with respect to this criterion, 50% or more statistically significant responders in each of the two groups (healthy and asthmatics) were found (see rows 1–4 in *Table 5* above). Notably, significant differences found between pre- and post-challenge time series were, in general, not attributable to changes in the variance, as verified using Levene's test (results not shown). Nevertheless, for most participants the lung function parameters did not show short-term/transient relative changes induced by the viral challenge that were statistically significantly different in magnitude from short-term changes observed during the pre-challenge phase (see columns 4 and 5, and rows 1–4 in *Table 5* above, and *Supplementary file 1*). Taken together, these results suggest that the changes in lung function elicited by the viral challenge are, both for healthy and asthmatic participants, subtle, spread over comparatively longer time periods, and unlike a transient decline. This is in line with the results of previous studies (*Seemungal et al., 2001*), which concluded that after RV challenge lung function in asthmatic subjects did change, but did not decline dramatically in comparison to the changes observed in healthy controls. In contrast, our analyses indicate that changes in the inflammatory or immunological biomarkers at the cellular or molecular level are short-term and transient in

**Table 6.** The demographics of the study population.
BMI is Body Mass Index. Only one healthy subject smoked two pack years or less 2 years before recruitment to our study, which is considered an insignificant smoking history. FEV1: forced expiratory volume in one second. PEF: peak expiratory flow.

| Demographic features | Healthy | Asthmatic |
|---|---|---|
| Total number, n | 12 | 12 |
| Female gender, n (%) | 7 (58.3%) | 8 (66.7%) |
| Age (years), mean (SD) | 21 ± 1.5 | 22.2 ± 2.2 |
| Ethnicity (Caucasian), n (non-Caucasian, n) | 11 | 9 |
| BMI, mean (SD) | 22.2 ± 1.6 | 22.8 ± 3.1 |
| Smoking (pack years), n | 1 (0.17 PY) | – |
| Height (centimeters) | 177.7 ± 8.6 | 172.5 ± 13.0 |
| Weight (KG) | 70.4 ± 10.1 | 67.8 ± 12.4 |
| Baseline spirometry | | |
| FEV1 %predicted | 105.7 ± 11.6 | 101.0 ± 10.0 |
| FVC %predicted | 104.2 ± 10.5 | 104.2 ± 10.2 |
| PEF %predicted | 108.4 ± 14.0 | 104.7 ± 12.2 |
| | | mean ± standard deviation |

**Table 7.** Basic characteristics of the study population.

| Healthy | Asthmatics |
| --- | --- |
| No history of episodic chest symptoms | History of episodic chest symptoms |
| Baseline FEV1 $\geq$ 80% predicted | Baseline FEV1 $\geq$ 70% predicted |
| AHR to methacholine ($PC_{20}$) $\geq$ 19.6 mg/ml | AHR to methacholine ($PC_{20}$) $\leq$ 9.8 mg/ml |
| SPT negative for all 12 common Aeroallergens | SPT positive for at least 1 out of 12 common Aeroallergens |

FEV1: forced expiratory volume in one second, AHR: Airway Hyper Responsiveness, $PC_{20}$: Provocative Concentration causing a 20% fall in FEV1, SPT: Skin Prick Test.

nature (see rows 5–8 in *Table 5* above, and *Supplementary file 1*). Nevertheless, for these parameters fewer responders were found, when compared to the lung function parameters. However, our results also hint at a relatively short time scale of response of these inflammatory/immunological biomarkers. Thus, the sampling frequency used in this study may not entirely capture the rapidly changing magnitudes. The observed differences in the type of response between the lung function and the inflammatory/immunological biomarkers may be a manifestation of the interplay of different temporal and spatial scales.

### Potential physiological and inflammatory mechanisms responsible for the biomarker dynamics observed in the group of asthmatics and the resulting reduction in adaptive capacity

One potential explanation for the differences in biomarker dynamics observed between healthy and asthmatic participants could be the level of airway obstruction. Indeed, Dames et al. quantitatively assessed the overall variability and complexity of airflow time series in patients with COPD during resting breathing. They found that airflow pattern complexity was reduced proportionally to airway obstruction measured with spirometric indices (*Dames et al., 2014*).

Changes at the level of airway smooth muscle (ASM) could also be associated with differences in the dynamics of system properties based on mechanical or environmental triggers affecting airway responsiveness (*Brook, 2014*; *Noble et al., 2012*). Lack of appropriate ASM-stretch has been associated with pathological incapacities of the asthmatic airways (*Slats et al., 2007*), which may mechanistically explain the inability to return back to homeokinetic range in asthma after a viral infection.

Inflammatory mediators and pathways involved in asthma could also explain the observed fluctuations in biomarker dynamics. With respect to the inflammatory biomarker dynamics it is well known that the production of inflammatory mediators is strictly controlled to ensure a limited but effective inflammatory response. To that end, most mRNAs encoding inflammatory mediators contain regulatory motifs like AU-rich elements in their 3'-untranslated region, which affect both transcriptional control, mRNA half-life and translational control (*Hao and Baltimore, 2009*). Changes in these regulatory mechanisms are likely to lead to less well-controlled inflammatory responses and in fact beyond that as the expression of many response genes are controlled by these regulatory motifs. We have recently shown that this translational control of AU-rich containing mRNAs in primary bronchial epithelial cells from mild and severe asthma patients compared to that from healthy controls is defective (*Ravi et al., 2019*). This led to exaggerated ex vivo production of, for example mediators driving neutrophilic inflammation, which correlated with the in vivo neutrophilic inflammation. In a follow-up study (A. Ravi et al, submitted) we showed that a RV16 challenge worsened this defect and with that the neutrophilic inflammation whereas this was not the case in healthy controls. The strong correlations between this defect and neutrophilic inflammation are suggestive of causality although formal proof is still lacking. This defect in the bronchial epithelial cells in asthma may underlie loss of adaptive capacity in response to a RV16 challenge.

### Limitations of the study

Our findings need to be judged in light of the limitations of our study. One of the limitations is that we only included mild asthmatics and therefore our findings may not be directly translatable to moderate and severe asthma. A similar study setting for severe asthma is not feasible because it is

ethically not acceptable to challenge those asthma patients with RV16 and in addition severe asthmatics are likely to be on corticosteroid treatment therefore introducing a confounding factor. Another possible shortcoming of this study is the relatively small sample size. However, this drawback is compensated for by the unprecedented high sampling frequency at which the participants were screened in our study.

## Conclusion and implications

We presented evidence supporting the notion that a chronic disease such as asthma may alter the properties of a homeokinetic physiologic system in a way that compromises its capacity to appropriately react to a possibly harmful environmental stimulus. This loss of adaptive capacity in the asthmatic lung may be understood as changes that render the system overly unstable (*Frey et al., 2011*). As a proof of concept, such changes in homeokinetic system properties would provide evidence supporting the idea that not only singular factors in isolation, but their interaction and also system properties, such as the interactions between their constituent parts, may contribute to disease dynamics and phenotype stability.

This systems-level understanding of chronic asthma may open up new avenues for better understanding of asthma and other chronic dynamic diseases. Already in this small sample size, it is obvious that there is remarkable individual temporal variability in inflammatory and physiological biomarkers, not only in disease but also in health. Thus, dynamic fluctuations of physiological processes around an equilibrium state, including their related biomarkers, are an intrinsic feature of the respiratory system. Moreover, even in health there are strong inter-individual differences in these dynamic characteristics. Nevertheless, within a given healthy subject, fluctuations remain similar following the virus challenge, indicating that, these dynamic fluctuations seem to be in a stable dynamic equilibrium state in health. This characteristic is lost in asthma.

Future studies involving time series of biomarker measurements may help us understand this system instability in chronic asthma or other airway diseases, even in the absence of severe airway inflammation or obstruction. Furthermore, future therapeutic approaches may want to focus on maintaining a stable homeokinetic equilibrium of the respiratory system, rather than just normalizing single physiological or inflammatory biomarkers.

## Materials and methods

This study was approved by the medical ethical committee from the Amsterdam University Medical Center and registered at the Netherlands Trial Register (NTR5426/NL5317).

### Participant cohort

Twelve non-smoking, atopic (as determined by positive skin prick test to common aeroallergens), mild to moderate asthmatic subjects (based on ATS/ERS criteria), not using steroids were chosen for inclusion. Similarly, 12 non-smoking, non-atopic healthy subjects were also included in the study as controls. All participants provided written informed consent. The demographics of the study population are summarized in *Table 6*. All the participants were required to have their serum antibody titer of RV16 <1:8 during screening. The age group for the study population was 18–30 years. Individuals with concomitant disease and pregnant women were excluded.

The basic inclusion criteria for the study populations followed standard recommendations and were as shown in *Table 7* below.

### Study design

The project represents a prospective observational, follow-up study including patients with asthma and healthy controls with an experimental RV intervention.

The study participants were recruited after meticulous screening of volunteers (as mentioned in Appendix). The study was mainly divided into two phases. Phase 1 (stable phase) consisted of 2 months where these subjects were followed up and sampled diligently every alternate day (3 times a week for most of the measurements) with constant frequency at the hospital clinic. After that they were subjected to a standardized nasal dose of RV inoculation in the laboratory and followed up at the same frequency for one additional month (also called Phase two or unstable phase). In total the

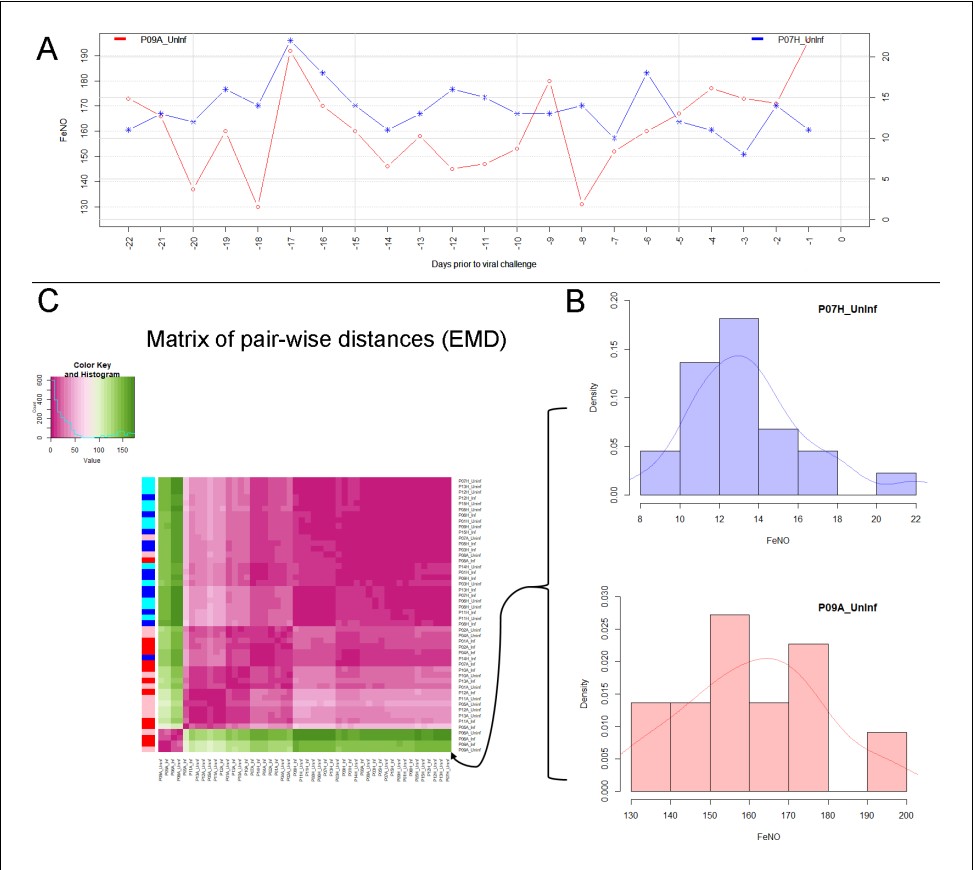

**Figure 2.** Analysis of biomarker time series in healthy and asthmatic populations using the "Earth Mover's Distance" metric. (**A**) Depicts two pre-challenge time series of FeNO obtained from a healthy (blue curve), and from an asthmatic (red curve) participant, respectively. (**B**) Each of the time series is represented as empirical distribution. This representation of the two time series allows for the calculation of a distance or 'dissimilarity' between the two by means of the Earth Mover's Distance (EMD). The EMD-comparison of all possible pairs of time series (both pre- and post-challenge) results in a symmetric matrix of pair-wise distances, as shown in (**C**) using a color-coded (violet to green) heat-map. Each row in this matrix corresponds to one time series. The color bar on the left hand side of the matrix encodes the 'type' of time series: Cyan marks a pre-challenge time series originating from a healthy participant; Blue marks a post-challenge time series originating from a healthy participant; Pink marks a pre-challenge time series originating from an asthmatic participant; Red marks a post-challenge time series originating from an asthmatic participant. The information stored in the matrix of pair-wise distances is then used within an agglomerative clustering algorithm in order to group the time series in different clusters. The outcome of this procedure is represented using a dendrogram as depicted in *Figure 1* above.

study consisted of 3 months of sampling period with a minimum of 180 measurements of lung function, 33 FeNO data points, and 20 cytokine and cell count measurements per subject.

The schematic work flow of the phases mentioned, is provided in the *Appendix 1—figure 15*.

## Measurement and collection of biomarkers
### Lung function assessment
Spirometry was performed only once on the screening visit at the clinic to include participants based on inclusion criteria using a daily calibrated spirometer according to European Respiratory Society (ERS) recommendations (*Miller et al., 2005*).

Home monitoring of morning and evening lung function was done by hand held devices (Micro Diary, CareFusion, yielding the FEV1, FVC, FEV1/FVC and PEF values analyzed in this study. Moreover, the Asthma Control Questionnaire was administered.

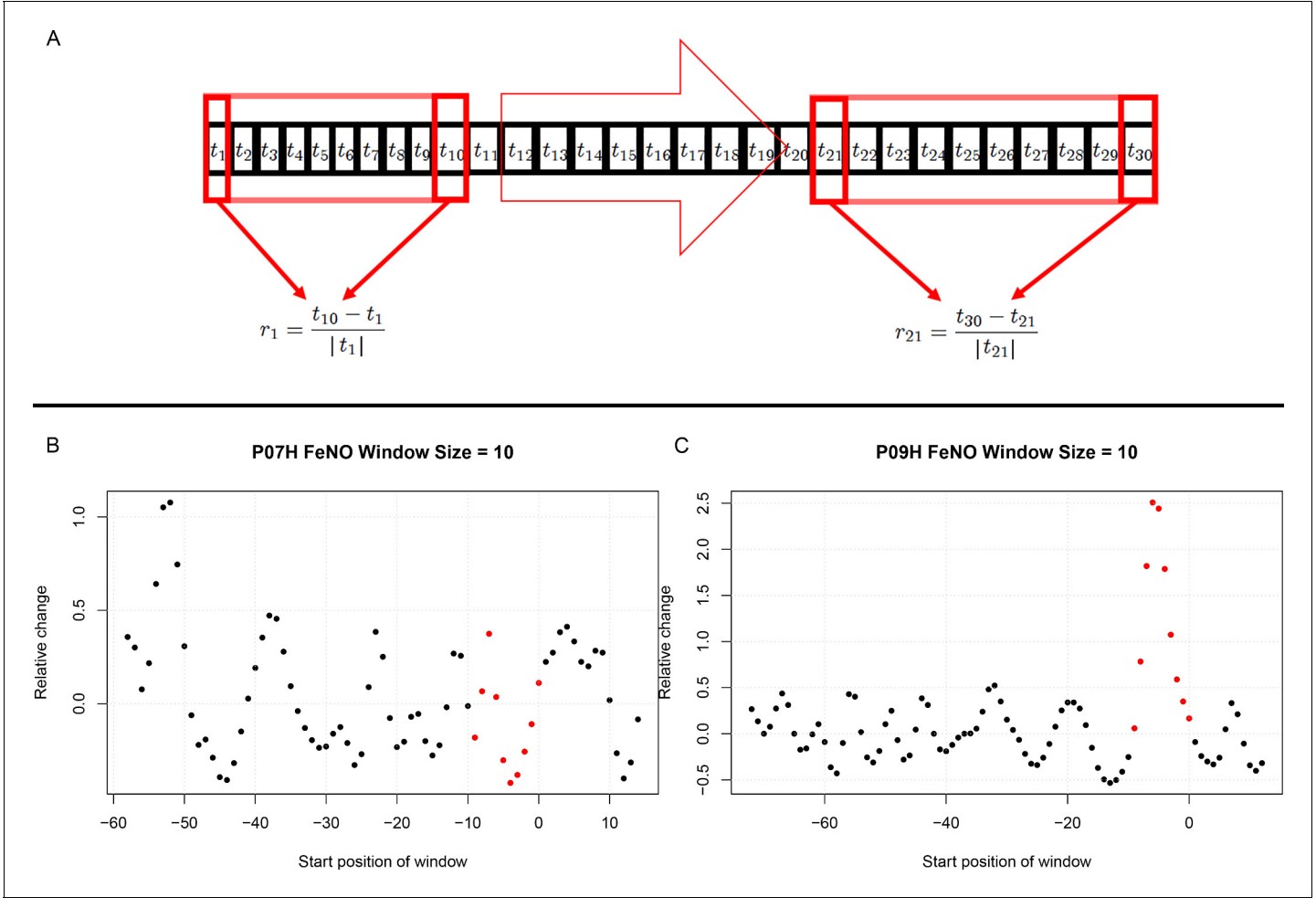

**Figure 3.** Estimation of short term responses in biomarker time series induced by viral challenge. (A) Graphical representation of a biomarker time series $t_i$. For the calculation of short-term/transient changes, a gliding interval or window is moved, one day at a time, along the time series. The relative change between the first and last entry of the gliding window is calculated, resulting in a new time series of short-term relative changes $r_i$. (B) A healthy participant's time series of short-term relative changes in FeNO is depicted. A gliding interval of size 10 days was used to calculate it from the participant's time series of FeNO measurements. The start position of the gliding window is expressed relative to the day of the viral challenge, which is marked as day 0. When the position of the gliding window was such that the day of the viral challenge was contained within the gliding window, the corresponding value of the relative change is marked in red. In order to assess the statistical significance of the short-term relative changes possibly elicited by the viral challenge, the relative change values located to the left of those marked in red were compared to the values marked in red by means of a Mann-Whitney-U-test. Visual inspection of the time series in B correctly suggests that the outcome of this test is not significant. The reason being that the relative changes within time intervals of 10 days observed prior to the viral challenge are comparable to changes observed within intervals of the same length containing the day of the viral challenge. (C) Depicting data from a different healthy participant, the situation is clearly different, as verified by a significant outcome of the corresponding Mann-Whitney-U-test. In such cases, the participant is called a 'responder' with respect to the 'relative change within 10 days criterion'.

## Exhaled Nitric Oxide (FeNO)

Measurement of fractionated exhaled nitric oxide (FENO) was performed using the NIOX MINO (Aerocrine AB, Sweden). Single measurements per person were recorded at the clinic, thrice weekly, according to recommendations by the ATS (**Dweik et al., 2011**).

## Nasal lavage

Nasal lavage was collected from the study participants once weekly before RV challenge and was up scaled to thrice weekly after the challenge at the clinic as previously described (**Grünberg et al., 2001**) [Refer to Appendix for details].

*Table 8* provides an overview of the different sample measurements along with their frequency before and after rhinovirus challenge.

## Rhinovirus challenge

The study participants were exposed to rhinovirus 16 (RV16) using a standardized and validated challenge approach, based on previous studies by ourselves and other groups (*Grünberg et al., 1999*). All participants were screened for the presence of respiratory viruses just before the challenge, to rule out a concomitant infection resulting in a cold (see Appendix for more details). Those participants with a positive outcome of this test were excluded from the study. An experimental RV16 infection was induced by using a relatively low-dose inoculum of 100 TCID50 (Tissue Culture Infective Dose determining the amount of virus required to cause cytopathy in 50% of the cells) to mimic a natural exposure. The study protocol along with the viral dose used and its safety have been approved by the institutional Medical Ethics Committee in Amsterdam University Medical Centre, the details of which have been included in Appendix. Data from our previous study show that a low dose is sufficient to induce mild cold-symptoms (*Mallia et al., 2006*). Furthermore, this low-dose inoculum previously resulted in a slight decrease of FEV1 (loss of asthma control) in asthmatic patients between day 4 and 6 after RV16 exposure, whereas no decrease has been observed in healthy controls (*Grünberg et al., 1999*).

Refer to Appendix for further details.

## Statistical and computational analysis

Statistical tests resulting in a p-value less or equal to 0.05 were regarded as significant.

### Assessment of differences: Pre- vs. post-viral-challenge

For each participant, their time series of a given biomarker prior to and after the viral challenge were compared. This comparison was based on the Kolmogorov-Smirnov test, whereby the time series were treated as empirical distributions, thus disregarding the chronological order of the measurements.

Differences in the variance between the pre- and post-challenge distributions were assessed using Levene's test (*Levene, 1961*).

Multiple comparison correction was performed where required, using the false discovery rate (FDR) method of *Benjamini and Hochberg (1995)*, setting the expected proportion of falsely rejected null hypotheses to 0.05.

**Table 8.** The overview of different measurements performed in the study along with the frequency of sampling before and after rhino-virus challenge.
Measures 1–4 include repeated measurements and 5,6 represent one-time measurement to screen the subjects for the study. eight refers to the experimental intervention in the study. FEV1: forced expiratory volume in one second. FVC: forced vital capacity. PEF: peak expiratory flow. FeNO: fractional expired concentration of nitric oxide.

| Measurements of biomarkers | Frequency before rhinovirus challenge | Frequency after rhinovirus challenge |
|---|---|---|
| Lung function with pocket-size spirometers (FEV1, FVC, FEV1/FVC, PEF) | 2x daily | 2x daily |
| Exhaled Nitric Oxide (FeNO) | 3x weekly | 3x weekly |
| Differential cell counts | 1x weekly | 3x weekly |
| Asthma Control Questionnaire | 2x daily | 2x daily |
| Spirometry | Performed once during screening to include subjects in the study | |
| Methacholine challenge | Performed once during screening to include subjects in the study | |
| Rhinovirus challenge | Performed after 2 months into the study | |

The time series of a given biomarker, prior to and after the viral challenge, were regarded as empirical distributions and compared to each other using the Earth Mover's Distance (EMD) (*Rubner et al., 1998*). The resulting pair-wise distances between distributions were then used for hierarchical clustering of pre- and post-challenge distributions. See *Figure 2* and the Appendix for more details. Our clustering approach makes use of the entire time series (distributions) of values measured before and after the challenge, respectively, and does not amalgamate the information into a single magnitude (e.g., the mean value). This method unveils subtle differences and similarities between the participants' measurements that are less likely to be captured by conventional methods based on averages.

## Calculation of short-term/transient changes

For each participant individually, and for each biomarker, throughout the entire period of observation, the biomarker's relative change in value taking place within time intervals of 10 days was calculated. This choice of time interval length was made based on published literature whereby 5 days post exposure to respiratory viruses was shown to be critical. Hence a 10 day window for comparison would include 5 days before challenge to contrast with 5 days after challenge (*Denlinger et al., 2011*). This was done throughout the entire period of observation considering all possible time intervals consisting of 10 consecutive days. In order to assess the statistical significance of the short-term relative changes possibly elicited by the viral challenge, the magnitude (that is, the absolute value) of relative changes observed during 10 day time intervals starting at least 10 days prior to the challenge were compared, by means of a Mann-Whitney-U-test, to the magnitude of relative changes that took place during 10 day time intervals that contained the day of the challenge. See *Figure 3* and the Appendix for more details.

## Characterization of the dendrogram clusters

In order to evaluate the discriminatory power of a given biomarker, the clusters found in the clustering dendrogram were tested for enrichment in or depletion of healthy or asthmatic participants, and/or for enrichment in or depletion of pre- or post-challenge distributions. Statistically significant enrichment or depletion were established using the hypergeometric test (*Rubner et al., 1998*).

The relative location of leaves in the clustering dendrogram was quantitatively evaluated using the cophenetic distance (*Sokal and Rohlf, 1962*). The cophenetic distance between two leaves of a dendrogram is defined as the height of the dendrogram at which the two largest branches that individually contain the two leaves merge into a single branch.

For every cohort participant and any given biomarker there is a pre-challenge and a post-challenge time series, which we call the participant's pre- and post-pair. If the disruption caused by the viral challenge is not strong enough, the pre- and post-challenge distributions of a given participant will tend to cluster together. Therefore, a cluster in which pre- and post-pairs are closely located in terms of the cophenetic distance within the dendrogram, represents a subgroup of participants for which the viral challenge caused a relatively weaker disruption, at least with respect to the biomarker under scrutiny.

Two dendrogram leaves are called neighbors if their mutual cophenetic distance is equal to the minimum of all cophenetic distances from one of the leaves to all the other leaves in the dendrogram. If this condition is fulfilled for both leaves simultaneously, then the two leaves form a two-element cluster in the dendrogram. If the condition is only fulfilled for one of the leaves, the two are still considered neighbors, even if this is not always visually obvious from inspecting the dendrogram (see *Appendix 1—figure 16*).

Under the null-hypothesis that the branching in the dendrogram is the result of a purely random process, the number of neighboring pre- and post-pairs to be expected just by chance within a given cluster can be estimated by simply permuting the labels of the leaves in the dendrogram and counting the number of neighboring pre- and post-pairs. This permutation test is used for calculating the empirical p-values displayed in *Tables 2* and *4* above.

A participant is fully represented in a given cluster if both their pre- and post-challenge time series of measurements are contained in the cluster. For example, the healthy participant 'P08H' is fully represented in Cluster 2, as both their pre- and post-challenge time series of FeNO measurements are members of Cluster 2 (see *Figure 1* above). Partial representation corresponds to the

scenario in which only one of the two time series (pre- and post-challenge) is a member of the cluster. For instance, the asthmatic participant " P07A' is only partially represented in Cluster 2, because their pre-challenge time series of FeNO measurements is part of Cluster 2, whereas their post-challenge time series of FeNO belongs to Cluster 3 (see *Figure 1* above).

### Autocorrelation of time series

The autocorrelation of a given time series was calculated using the sample Pearson correlation coefficient of the original time series and the time series resulting after forward-shifting the original time series by the lag utilized. In other words, if the original time series consists of the values $x_1, \ldots, x_n$ and $L$ is the lag, all complete (i.e., no member of the pair is a missing value) pairs $(x_t, x_{t-L})$ for $t = L+1, \ldots, n$ are used to calculate the covariance in the formula of the sample Pearson correlation coefficient.

## Acknowledgements

The salary of AS was sponsored from the European Respiratory Society-Marie Sklodowska Curie actions COFUND RESPIRE two fellowships (MCF-7077–2014) and also from a grant supported by Swiss Lung Association (Lungenliga Schweiz) (2017_14). The work was supported by an unrestricted grant from Chiesi Pharmaceuticals, institutional funding from the Academic Medical Centre, Amsterdam UMC, University of Amsterdam (IA601011).

The authors would like to thank Dr. Sven Schulzke, Dr. Michael Shapiro, and Dr. Florian Geier for their very valuable feedback on initial versions of this manuscript.

## Additional information

### Funding

| Funder | Grant reference number | Author |
|---|---|---|
| European Respiratory Society | Respire 2 (MCF-7077-2014) | Anirban Sinha |
| Chiesi Pharmaceuticals | | Anirban Sinha |
| H2020 Marie Skłodowska-Curie Actions | Respire 2 (MCF-7077-2014) | Anirban Sinha |
| Swiss Lung Association | 2017_14 | Anirban Sinha |

The funders had no role in study design, data collection and interpretation, or the decision to submit the work for publication.

### Author contributions

Anirban Sinha, Conceptualization, Data curation, Formal analysis, Funding acquisition, Validation, Investigation, Methodology, Writing—original draft, Project administration, Writing—review and editing; René Lutter, Conceptualization, Supervision, Investigation, Writing—original draft, Project administration, Writing—review and editing; Binbin Xu, Data curation, Formal analysis, Visualization; Tamara Dekker, Barbara Dierdorp, Formal analysis, Methodology; Peter J Sterk, Conceptualization, Supervision, Funding acquisition, Writing—original draft, Project administration, Writing—review and editing; Urs Frey, Conceptualization, Funding acquisition, Writing—original draft, Project administration, Writing—review and editing; Edgar Delgado Eckert, Conceptualization, Data curation, Formal analysis, Supervision, Validation, Methodology, Writing—original draft, Writing—review and editing

### Author ORCIDs

Anirban Sinha ![ORCID] https://orcid.org/0000-0002-4146-9687
Edgar Delgado Eckert ![ORCID] https://orcid.org/0000-0001-6415-4971

## Ethics

Human subjects: The study protocol along with the viral dose used and its safety have been approved by the institutional Medical Ethics Committee in Amsterdam University Medical Centre (Protocol No. NL54293.018.15). The trial has been registered at the Netherlands Trial Register (Netherlands Trial Register (NTR5426/NL5317). Proper Informed Consent was taken from every participant before including them in the study.

## Decision letter and Author response

Decision letter https://doi.org/10.7554/eLife.47969.sa1
Author response https://doi.org/10.7554/eLife.47969.sa2

## Additional files

### Supplementary files

• Supplementary file 1. Plots of the time series of relative changes within 10 days for each biomarker.

• Supplementary file 2. Plots of the time series of each biomarker.

• Transparent reporting form

### Data availability

All data generated or analysed during this study are included in the manuscript and supporting files.

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

# Appendix 1

## Results

### Effectiveness of the viral inoculation

Each participant in the study was administered the same dose of the virus (100 TCD 50) through the nose and every subject was tested for being positive for the virus after inoculation. False positive results due to previous exposure to the virus was ruled out by strict inclusion criteria of not having the titer of antibodies against RV16 >1:8 in serum, measured at screening and prior to inoculation.

Positivity to viral inoculation was confirmed by either one of these three criteria

1. Positive test for antibodies against RV at the terminal visits of the participant
2. Positive RV PCR test from approximately 3$^{rd}$ day (second visit) post RV challenge
3. Symptoms of RV induced cold

The effect of the virus in every study participant is summarized below in *Table 1* (Column one indicates the seroconversion of the antibodies against the virus at the end of the study, Column two reflects the response to the virus by the PCR product on the day three after challenge and finally Column three shows the symptoms developed in the volunteers after the viral inoculation). Positive response in either of the three categories was considered evidence for a successful viral challenge.

**Appendix 1—table 1.** Effectiveness of the viral challenge in asthmatic and healthy participants, respectively.
A participant is considered to have a successful viral inoculation if any one of the three tests is positive. one indicates positive response and 0 indicates a failed response in the corresponding tests indicated in columns.

| Study volunteers | Responders by seroconversion | Responders by RVPCR | Description of symptoms |
|---|---|---|---|
| 01A | 1 | 1 | Running nose, blocked nose and cough |
| 02A | 0 | 1 | Sore throat, blocked nose, full head, coughing and sneezing. |
| 04A | 1 | 0 | Slight symptoms of cold, very mild |
| 05A | 0 | 1 | No clear symptoms |
| 06A | 1 | 1 | Symptoms of cold |
| 07A | 1 | 1 | Minor sore throat, dripping nose, shortness of breath. |
| 08A | 1 | 0 | Sore throat, Probably very mild effect, no other symptoms |
| 09A | 0 | 1 | Running nose, head ache, fever |
| 10A | 1 | 1 | Very mild symptoms |
| 11A | 1 | 1 | Very mild symptoms |
| 12A | 0 | 1 | Cough, blocked nose |
| 13A | 0 | 1 | No clear symptoms |
| 01H | 0 | 1 | Sore throat |
| 03H | 0 | 1 | Sore throat and blocked nose |
| 05H | 1 | 1 | Cough, blocked nose and headache |
| 06H | 0 | 1 | No clear symptoms observed |
| 07H | 1 | 0 | Sneezing, itchy eyes and a little bit of a cough |
| 08H | 1 | 1 | Little bit sore throat |

*Continued on next page*

*Appendix 1—table 1 continued*

| Study volunteers | Responders by seroconversion | Responders by RVPCR | Description of symptoms |
| --- | --- | --- | --- |
| 09H | 1 | 1 | Sore throat and little cough |
| 11H | 0 | 0 | Blocked nose and sputum |
| 12H | 1 | 1 | Running nose and sneeze |
| 13H | 1 | 1 | No clear symptoms |
| 14H | 0 | 1 | Cough and blocked nose |
| 15H | 1 | 0 | No clear symptoms |

## Clustering based on time series of biomarkers
Percentage of eosinophils in nasal lavage fluid

The corresponding dendrogram is depicted in *Appendix 1—figure 1* below.

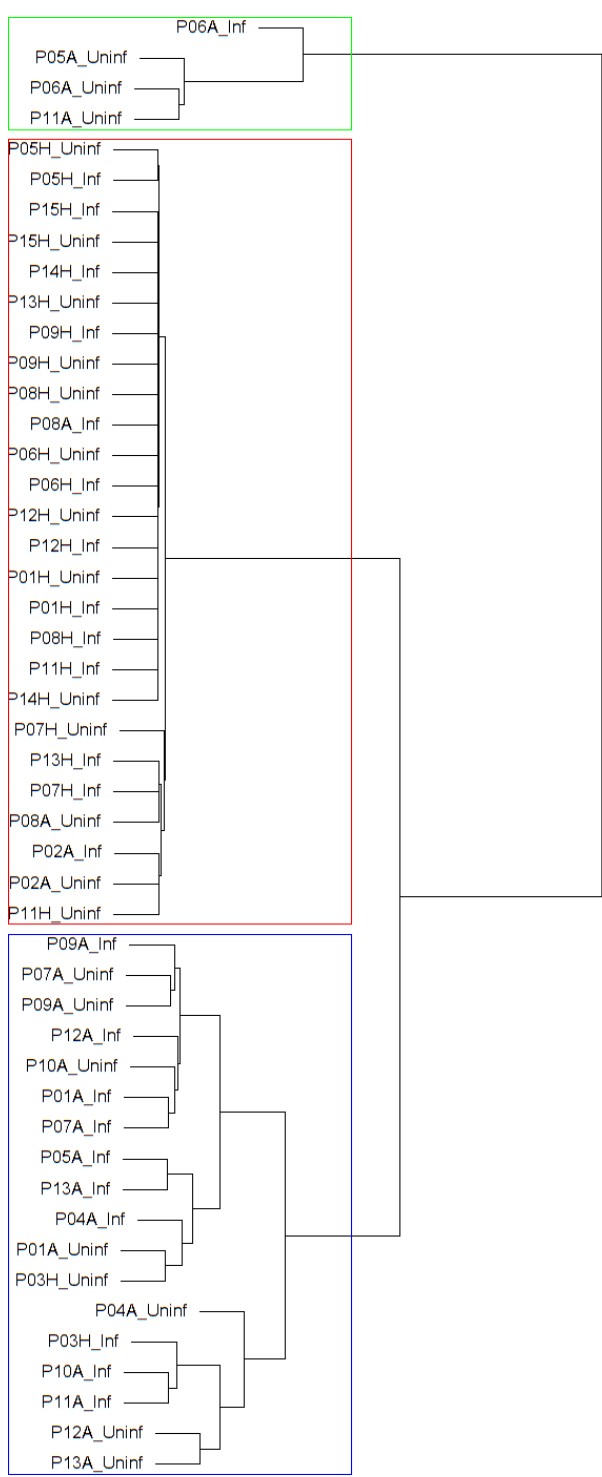

**Appendix 1—figure 1.** Dendrogram obtained from clustering the participants' time series of the percentage of eosinophils in nasal lavage fluid using the EMD.

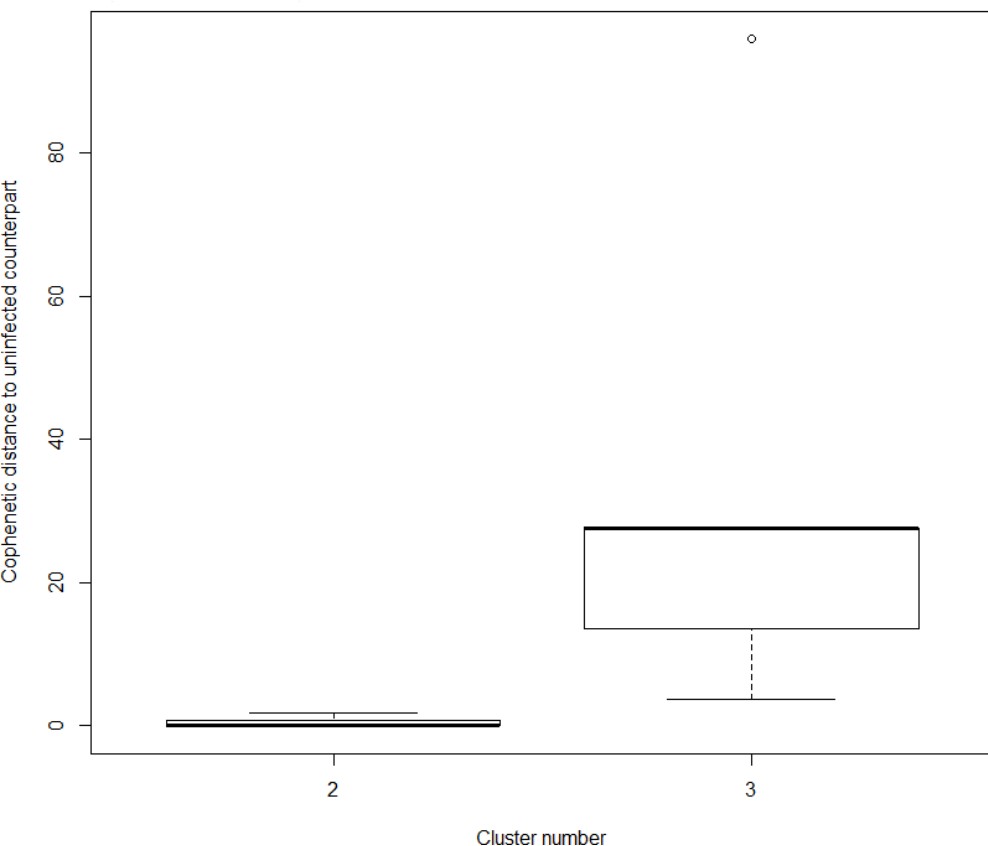

**Appendix 1—figure 2.** The boxplot to the left represents the distribution of cophenetic distances between time series corresponding to infected healthy participants and their uninfected counterparts. Only time series belonging to Cluster two in the clustering dendrogram obtained using the percentage of eosinophils in nasal lavage fluid (see *Appendix 1—figure 1* above) are contemplated here. The boxplot to the right represents the distribution of cophenetic distances between time series corresponding to infected asthmatic participants and their uninfected counterparts. Only time series belonging to Cluster three in the clustering dendrogram obtained using the percentage of eosinophils in nasal lavage fluid (see *Appendix 1—figure 1* above) are contemplated here. The two distributions are statistically significantly different (p-value=8.96e-05, one-tailed Mann-Whitney-U-test, the cophenetic distances in Cluster three being, on average, higher than the ones in Cluster 2.

## FeNO (Exhaled Nitric Oxide)

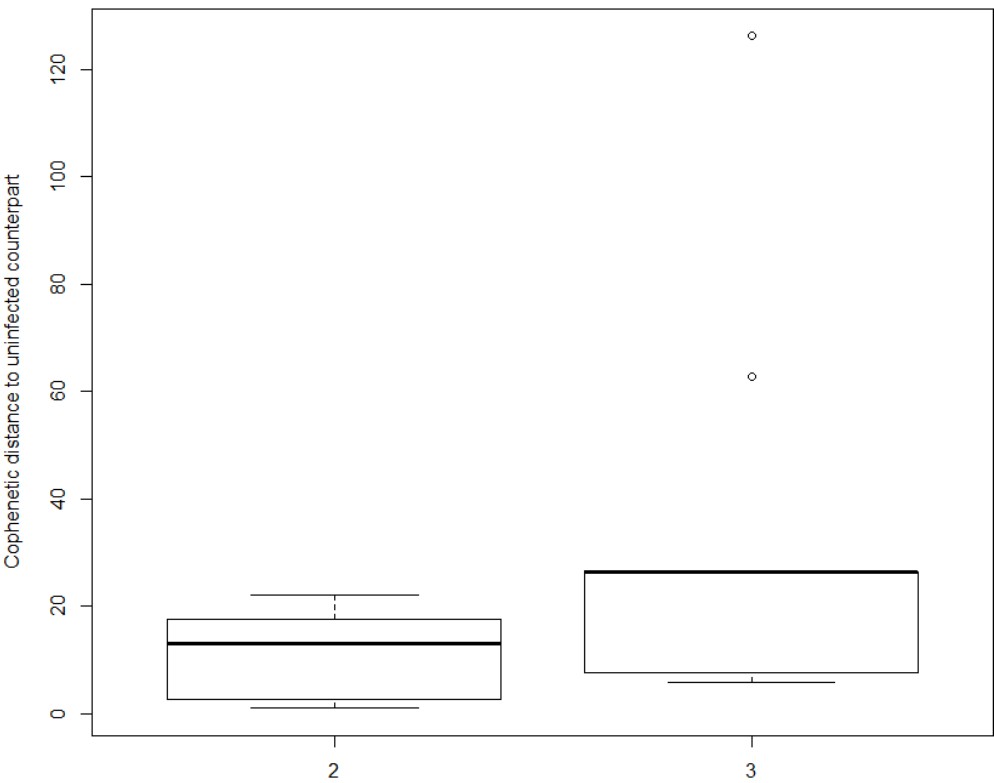

**Appendix 1—figure 3.** The boxplot to the left represents the distribution of cophenetic distances between time series corresponding to infected healthy participants and their uninfected counterparts. Only time series belonging to Cluster two in the clustering dendrogram obtained using FeNo data (see *Figure 1* in the Main Manuscript) are contemplated here. The boxplot to the right represents the distribution of cophenetic distances between time series corresponding to infected asthmatic participants and their uninfected counterparts. Only time series belonging to Cluster three in the clustering dendrogram obtained using FeNo data (see *Figure 1* in the Main Manuscript) are contemplated here. The two distributions are statistically significantly different (p-value=0.033, one-tailed Mann-Whitney-U-test, the cophenetic distances in Cluster three being, on average, higher than the ones in Cluster 2.

## Cell density in nasal lavage fluid

We found one cluster of size 25 enriched in pre-challenge time series (p=0.041), and one cluster of size seven enriched in asthmatics (p=0.049). See *Appendix 1—figure 2*

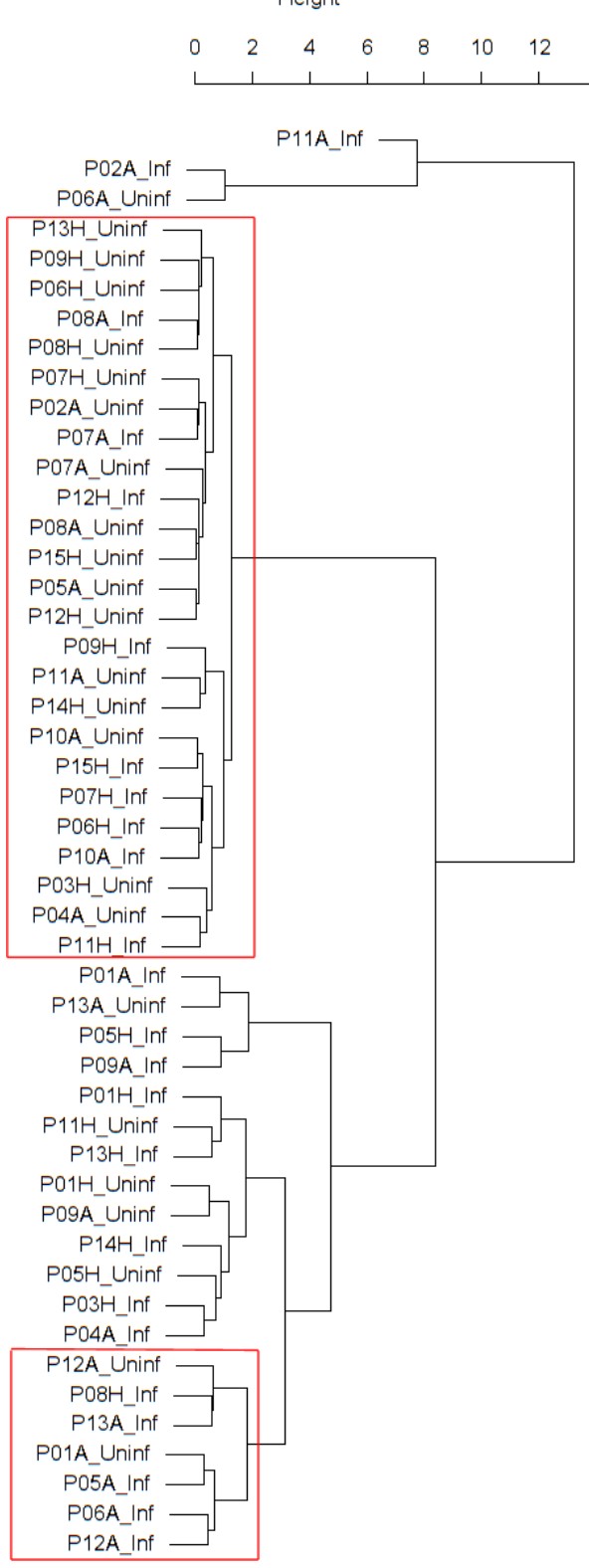

**Appendix 1—figure 4.** Dendrogram obtained from clustering the participants' time series of cell density (millions per ml) in nasal lavage fluid using the EMD.

## Percentage of neutrophils in nasal lavage fluid

We found one cluster of size 10 enriched in time series from healthy participants (p=0.036). More-over, no clusters were found showing a significant enrichment in or depletion of pre-challenge or post-challenge time series.

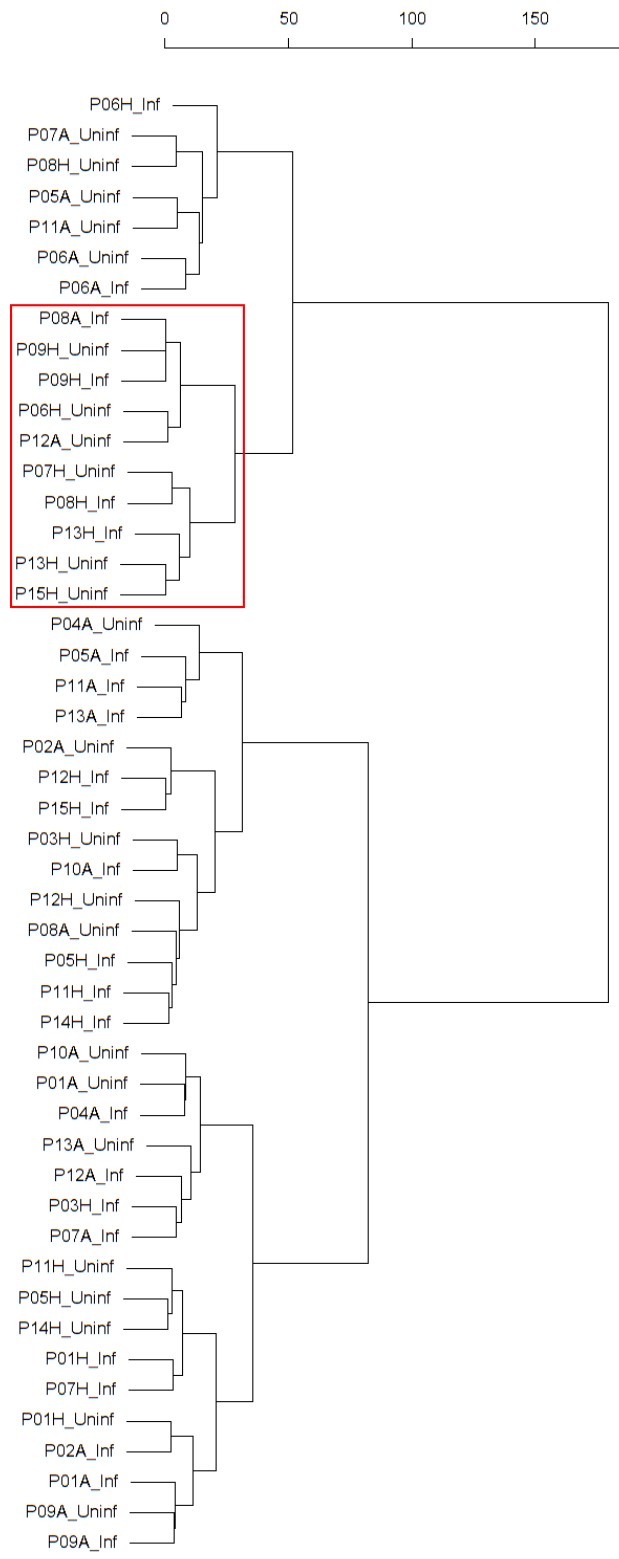

**Appendix 1—figure 5.** Dendrogram obtained from clustering the participants' time series of the percentage of neutrophils in nasal lavage fluid using the EMD.

## Lung function parameters

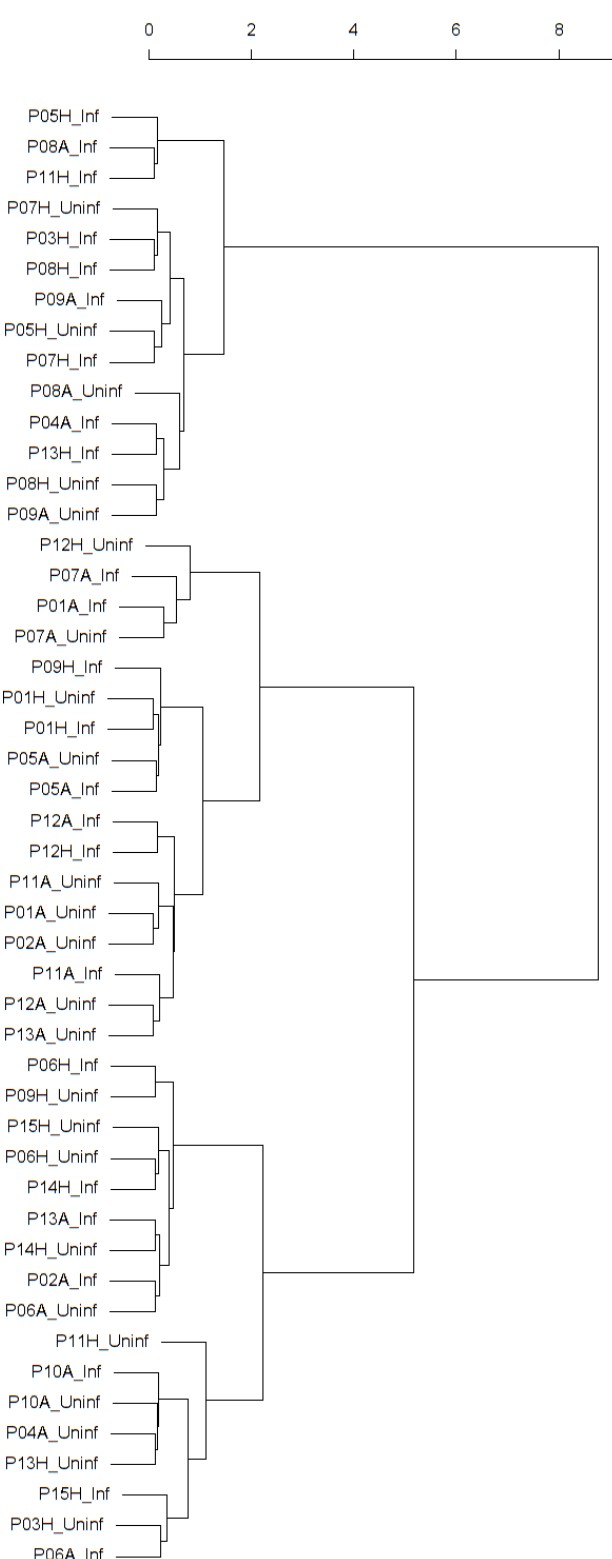

**Appendix 1—figure 6.** Dendrogram obtained from clustering the participants' time series of the normalized ratio FEV1/FVC using the EMD.

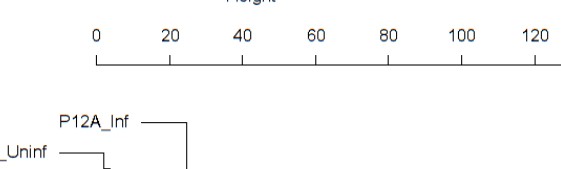

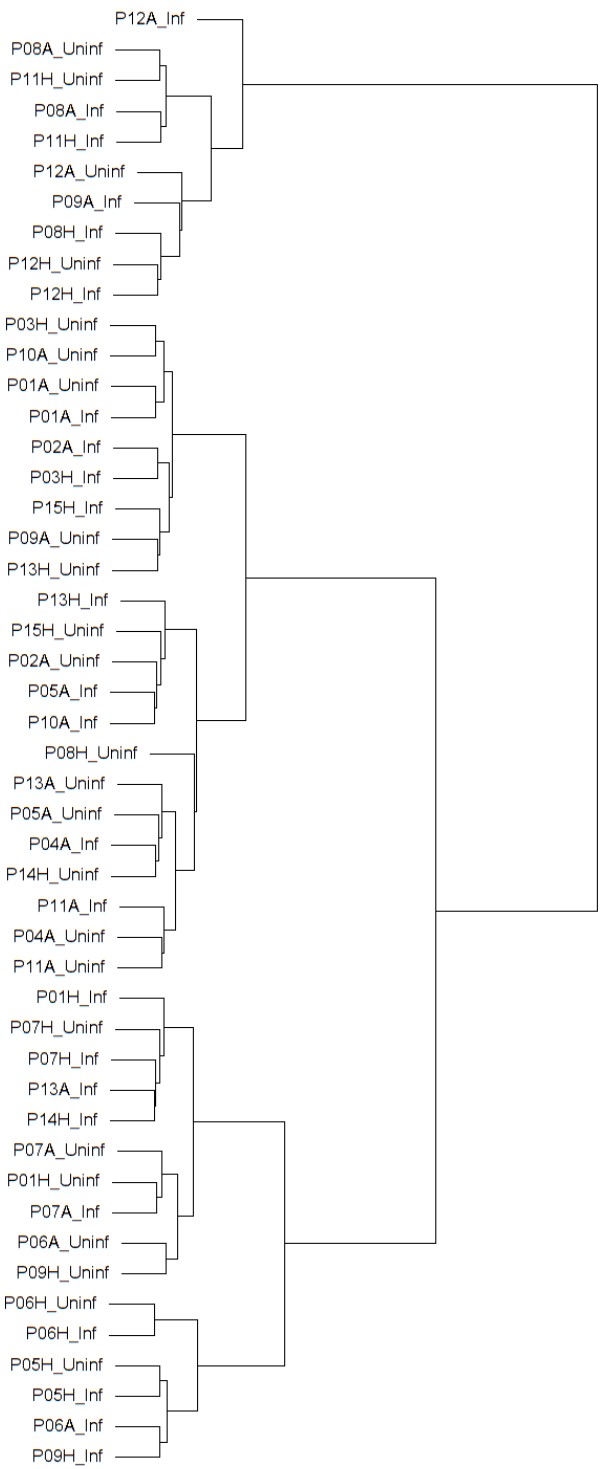

**Appendix 1—figure 7.** Dendrogram obtained from clustering the participants' time series of PEF (% predicted) using the EMD.

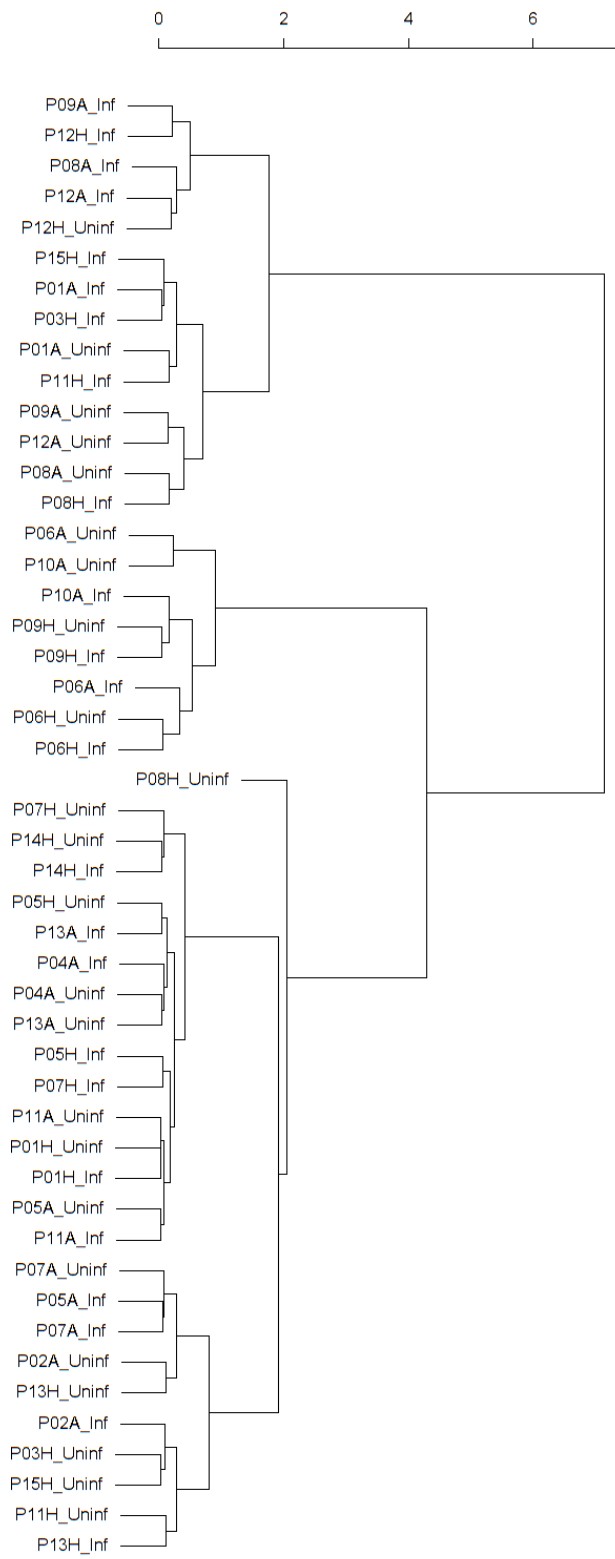

**Appendix 1—figure 8.** Dendrogram obtained from clustering the participants' time series of normalized FEV1 using the EMD.

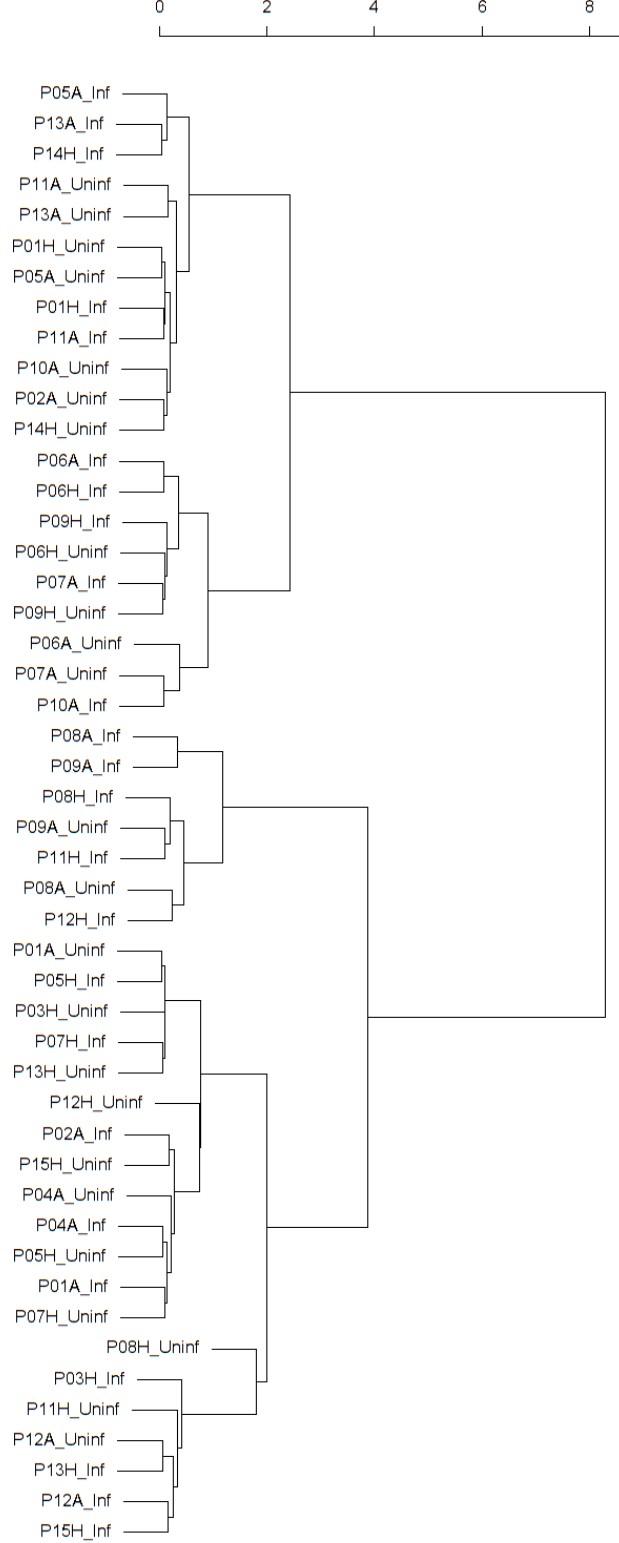

**Appendix 1—figure 9.** Dendrogram obtained from clustering the participants' time series of normalized FVC using the EMD.

## Sensitivity analysis of main clustering results

We investigated the sensitivity of the clustering of FeNO time series to changes in the data via non-parametric bootstrapping. However, given that the post-viral challenge time series are very short (eleven data points or fewer), resampling would be strongly affected by small sample size effects (*Isaksson et al., 2008*). Thus, we only applied bootstrapping to the pre-challenge time series. Moreover, we resorted to soft bootstrapping (*Mucha and Bartel, 2014*) (see Materials and methods below for more details) in order to increase the likelihood that least frequent values in the time series would be chosen during the resampling procedure. The results from 1000 soft bootstrapping iterations were as follows:

In 100% of the resulting soft bootstrap dendrograms, Cluster 1 ('outliers' cluster) was found. Moreover, two additional clusters, one, Cluster 2', significantly enriched in time series stemming from heathy participants, and another, Cluster 3', significantly enriched in time series stemming from asthmatic participants were found in 100% of the resulting soft bootstrap dendrograms. In other words, there was always a bootstrap counterpart to clusters 1,2, and three as found in the dendrogram obtained using the original, unperturbed data (see *Figure 1* in the Main Manuscript).

In 51.4% of the soft bootstrap dendrograms, Cluster 3' (the cluster enriched in time series stemming from asthmatic participants) contained a subcluster enriched in post-challenge time series. Whereas, only in 7.9% of the soft bootstrap dendrograms, Cluster 2' (the cluster enriched in time series stemming from healthy participants) contained a subcluster enriched in post-challenge time series. (cf. *Table 3* in the Main Manuscript).

Bootstrap distribution of mean cophenetic distances between the members of all pre- and post-pairs contained in Cluster 2':

Min. 1 st Qu. **Median** Mean 3rd Qu. Max.

6.358 11.017 **11.954** 13.527 14.947 26.654

Bootstrap distribution of mean cophenetic distances between the members of all pre- and post-pairs contained in Cluster 3':

Min. 1 st Qu. **Median** Mean 3rd Qu. Max.

7.036 21.581 **33.691** 29.321 35.610 49.080

In 80.6% of the soft bootstrap dendrograms, the mean cophenetic distances between the members of all pre- and post-pairs contained in Cluster 2' was smaller than the mean cophenetic distances between the members of all pre- and post-pairs contained in Cluster 3'.

Bootstrap distribution of the p-values resulting from the one-tailed Mann-Whitney-U-test comparing the distribution of cophenetic distances between time series corresponding to infected healthy participants and their uninfected counterparts in Cluster 2', to the distribution of cophenetic distances between time series corresponding to infected asthmatic participants and their uninfected counterparts in Cluster 3' (cf. *Appendix 1—figure 3*):

Min. 1 st Qu. **Median** Mean 3rd Qu. Max.

0.0005046 0.0163020 **0.0397435** 0.1563541 0.2208664 0.9537708

In 54.2% of the soft bootstrap dendrograms the resulting p-value was smaller or equal to 0.05.

Bootstrap distribution of the percentage of neighboring pre- and post-pairs in Cluster 2':

Min. 1 st Qu. **Median** Mean 3rd Qu. Max.

0.00 14.29 **19.90** 19.13 23.53 42.86

Bootstrap distribution of the percentage of neighboring pre- and post-pairs in Cluster 3':

Min. 1 st Qu. **Median** Mean 3rd Qu. Max.

8.33 20.00 **25.00** 28.63 37.50 80.00

In 27.1% of the soft bootstrap dendrograms, the percentage of neighboring pre- and post-pairs in Cluster 2'was bigger than the percentage of neighboring pre- and post-pairs in Cluster 3' (cf. *Table 2* in the Main Manuscript).

Bootstrap distribution of the empirical p-values resulting from the permutation test used for establishing the statistical significance of the proportion of neighboring pre- and post-pairs found in Cluster 2':

Min. 1 st Qu. **Median** Mean 3rd Qu. Max.

0.00000 0.00596 **0.03780** 0.12118 0.15008 1.00000

In 52.7% of the soft bootstrap dendrograms, the resulting empirical p-value was smaller or equal to 0.05.

Bootstrap distribution of the empirical p-values resulting from the permutation test used for establishing the statistical significance of the proportion of neighboring pre- and post-pairs found in Cluster 3':

Min. 1 st Qu. **Median** Mean 3rd Qu. Max.

0.000000 0.004287 **0.014640** 0.080993 0.080477 0.450480

In 57.5% of the soft bootstrap dendrograms, the resulting empirical p-value was smaller or equal to 0.05.

Soft bootstrapping of the time series of the percentage of eosinophils in nasal lavage fluid yielded comparable results (data not shown).

## Autocorrelation analysis of time series of biomarkers

For every participant, the autocorrelation coefficient of the lung function parameters time series and of the FeNO time series was calculated using a for each parameter type physiologically meaningful time lag. More specifically, a one-day lag was used for lung function parameters, and a two-days lag for FeNO. Due to the low sampling frequency used before the challenge, the time series of eosinophil and neutrophil cell density in nasal lavage fluid were not included in the autocorrelation analysis.

The autocorrelation of a given time series was calculated using the sample Pearson correlation coefficient of the original time series and the time series resulting after forward-shifting the original time series by the lag utilized.

The figures below (S10-14) display the distribution of the autocorrelation coefficient for each biomarker compared (FEV1, FEV1/FVC, FVC, PEF and FeNO) in the two groups before and after the viral challenge.

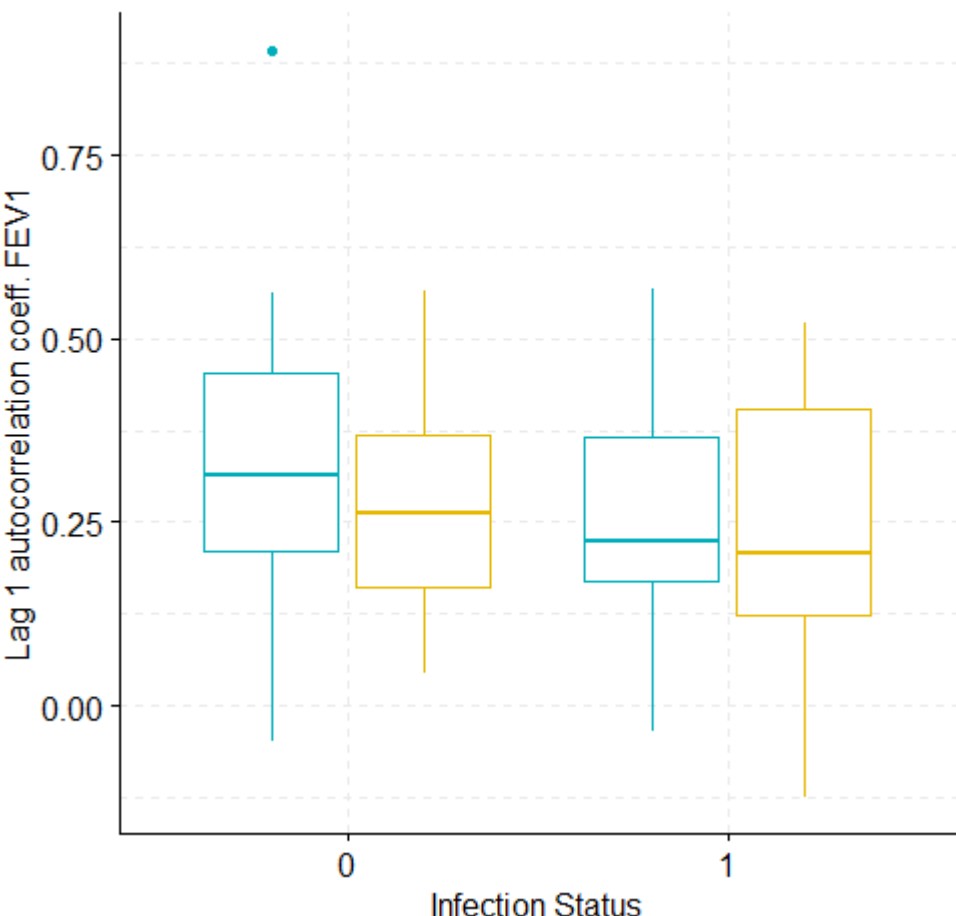

**Appendix 1—figure 10.** Boxplots demonstrating the distribution of autocorrelation coefficient at 1 day lag for FEV1 before and after viral challenge in healthy and asthma groups. A 2-way ANOVA resulted in no significant differences.

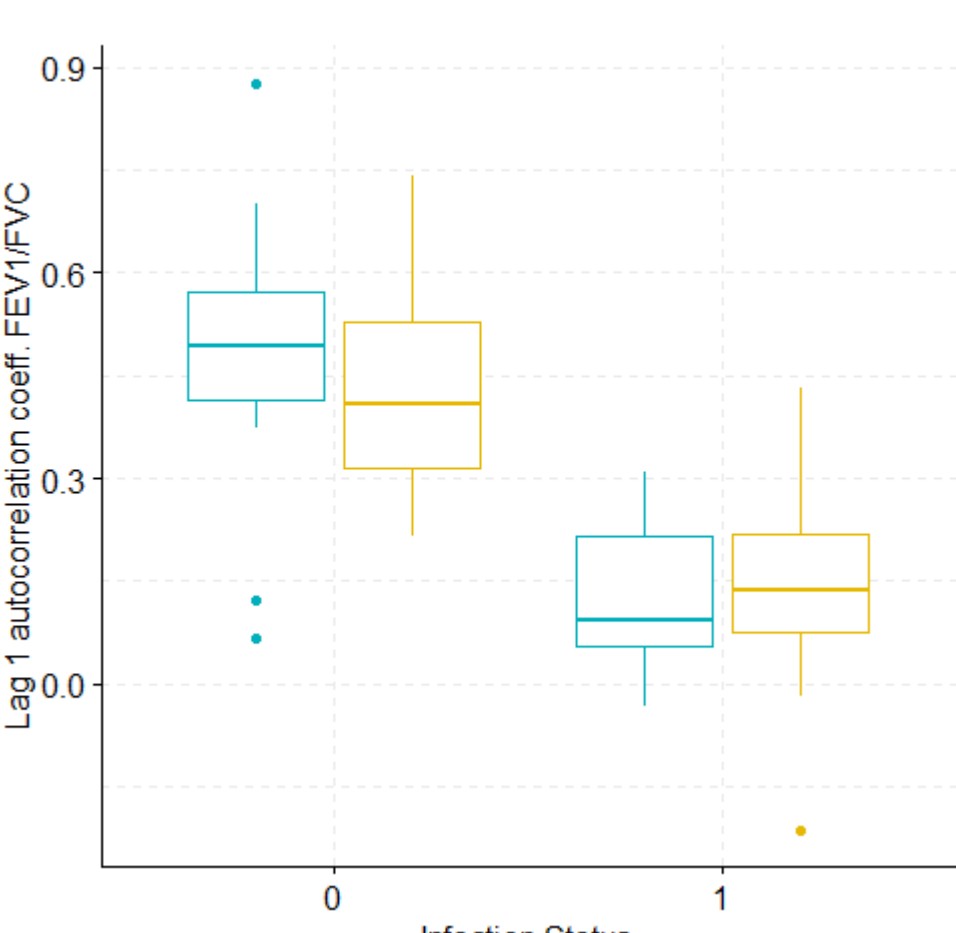

**Appendix 1—figure 11.** Boxplots demonstrating the distribution of autocorrelation coefficient at 1 day lag for FEV1/FVC before and after viral challenge in healthy and asthma groups. A 2-way ANOVA resulted in a significant impact of the infection status on the autocorrelation coefficient (p<1e-07).

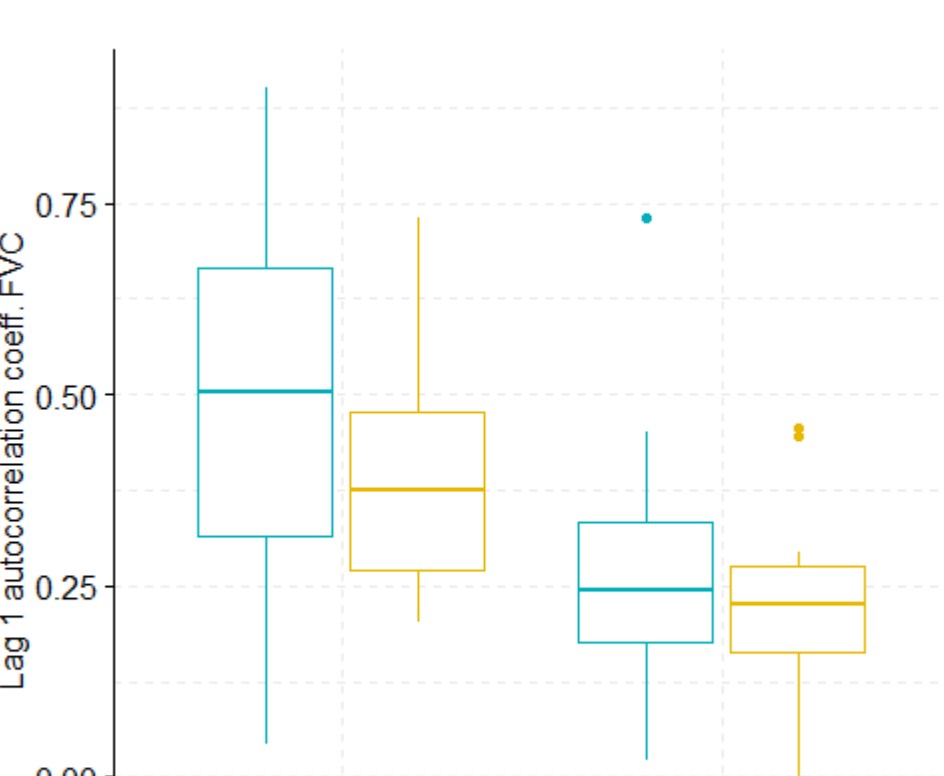

**Appendix 1—figure 12.** Boxplots demonstrating the distribution of autocorrelation coefficient at 1 day lag for FVC before and after viral challenge in healthy and asthma groups. A 2-way ANOVA resulted in a significant impact of the infection status on the autocorrelation coefficient (p=0.0007).

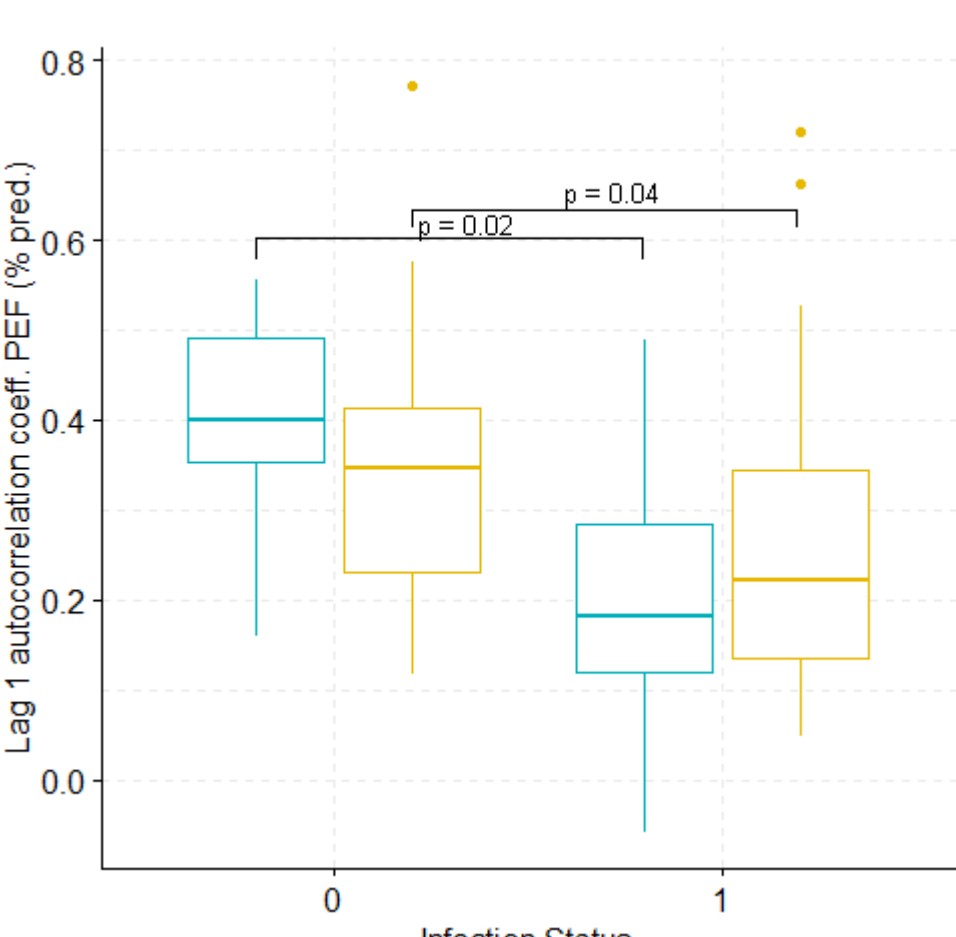

**Appendix 1—figure 13.** Boxplots demonstrating the distribution of autocorrelation coefficient at 1 day lag for PEF (% predicted) before and after viral challenge in healthy and asthma groups. The data did not fulfil the conditions for a 2-way ANOVA. Pairwise comparisons were carried out (t-test or Mann-Whitney-test, depending on whether the data fulfilled the conditions for a t-test). The tests used to compare pre- vs. post-challenge status within the two groups (healthy and asthma) were paired tests. Only significant p-values without multiple pairwise-comparison correction are displayed.

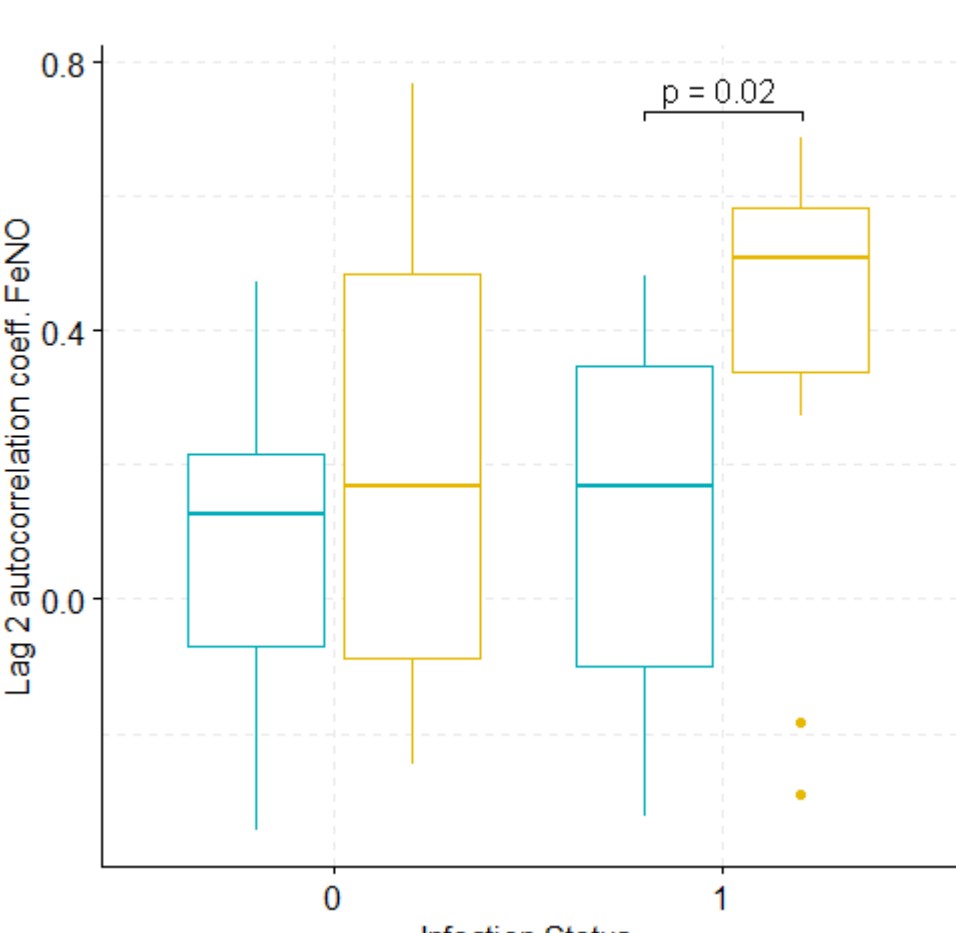

**Appendix 1—figure 14.** Boxplots demonstrating the distribution of autocorrelation coefficient at 2 days lag for FeNO before and after viral challenge in healthy and asthma groups. The data did not fulfil the conditions for a 2-way ANOVA. Pairwise comparisons were carried out (t-test or Mann-Whitney-test, depending on whether the data fulfilled the conditions for a t-test). The tests used to compare pre- vs. post-challenge status within the two groups (healthy and asthma) were paired tests. Only significant p-values without multiple pairwise-comparison correction are displayed.

**Appendix 1—table 2.** P-values of the group comparisons regarding the Pearson autocorrelation coefficient at respective lags of the biomarker time series as listed in the first column.

When the data did not fulfil the requirements for a 2-way ANOVA, pairwise comparisons were conducted followed by correction for multiple testing using the FDR method.

| Biomarker | Lag | P-values asthmatic vs. healthy participants prechallenge | P-values asthmatic vs. healthy participants postchallenge | P-values prechallenge vs. postchallenge in healthy participants | P-values prechallenge vs. postchallenge in asthmatics | P-values interaction (2-way ANOVA) |
|---|---|---|---|---|---|---|
| Normalized FEV1 | 1 day | 0.4431 | 0.4431 | 0.282 | 0.282 | 0.737 |
| Normalized FEV1/FVC | 1 day | 0.8664 | 0.8664 | <1e-07 | <1e-07 | 0.5439 |
| Normalized FVC | 1 day | 0.233 | 0.233 | 0.0007 | 0.0007 | 0.8475 |
| PEF (% pred.) | 1 day | 0.5062 | 0.5036 | 0.0698 | 0.085 | NA |
| FeNO | 2 days | 0.7349 | 0.082 | 0.822 | 0.4668 | NA |

The table below summarizes the results of comparing, in terms of the autocorrelation coefficient, the groups of asthmatic and healthy participants prior to and after the viral challenge. This comparison was conducted using a two-way ANOVA, when the data fulfilled the statistical requirements for the conduction of such a test. The latter requirements are normality (tested for using the Shapiro test) and homoscedasticity (tested for using Levene's test). For those biomarkers or derived magnitudes that did not fulfill the condition of homoscedasticity, the groups were compared pair-wise using the t-test or the Mann-Whitney-Wilcoxon test, depending on whether normality and homoscedasticity were given or not. Moreover, the comparison between pre- and post-challenge groups was conducted using the paired versions of the aforementioned tests. When conducting all four pairwise comparisons, p-values were corrected for multiple testing using the FDR method.

The tables below display each participant's individual pre- and post-challenge autocorrelation coefficient of the lung function parameters time series and of the FeNO time series. The accompanying p-values are the outcome of permutation tests aiming at assessing the statistical significance of the autocorrelation coefficients found. Patient IDs are indicated by Pxy, their health status using H/A, denoting Healthy or Asthmatic.

**Appendix 1—table 3.** Table displaying individual pre (0) and post (1) challenge autocorrelation coefficient at 1 day lag for FEV1 time series and associated p-values for the permutation tests conducted for assessment of the statistical significance of autocorrelation coefficients.

| Patient ID | Infection Status | 1 Day Lag FEV1 Autocorrelation | p-value |
|---|---|---|---|
| P01H | 0 | 0.254459145 | 0.006506 |
| P03H | 0 | 0.312001154 | 0.001425 |
| P05H | 0 | 0.382177632 | 0.000269 |
| P06H | 0 | 0.211190711 | 0.012607 |
| P07H | 0 | 0.203643391 | 0.03083 |
| P08H | 0 | 0.892125011 | <1e-6 |
| P09H | 0 | 0.318646957 | 0.000124 |
| P11H | 0 | 0.468008235 | <1e-6 |
| P12H | 0 | 0.562942585 | <1e-6 |
| P13H | 0 | −0.04987845 | 0.624876 |
| P14H | 0 | 0.447072337 | 6.00E-06 |

*Continued on next page*

*Appendix 1—table 3 continued*

| Patient ID | Infection Status | 1 Day Lag FEV1 Autocorrelation | p-value |
|---|---|---|---|
| P15H | 0 | 0.07979159 | 0.40799 |
| P01H | 1 | −0.035719445 | 0.82705 |
| P03H | 1 | 0.210750544 | 0.168883 |
| P05H | 1 | 0.408335039 | 0.009061 |
| P06H | 1 | 0.170326229 | 0.235598 |
| P07H | 1 | 0.166055078 | 0.245054 |
| P08H | 1 | 0.351014932 | 0.01394 |
| P09H | 1 | 0.464704837 | 0.001356 |
| P11H | 1 | 0.118074713 | 0.441357 |
| P12H | 1 | 0.207205794 | 0.181322 |
| P13H | 1 | 0.23673266 | 0.154677 |
| P14H | 1 | 0.56839312 | 2.50E-05 |
| P15H | 1 | 0.266373414 | 0.083097 |
| P01A | 0 | 0.200343624 | 0.062003 |
| P02A | 0 | 0.563837497 | <1e-6 |
| P04A | 0 | 0.239868384 | 0.028328 |
| P05A | 0 | 0.324114603 | 0.000172 |
| P06A | 0 | 0.348780303 | 0.000185 |
| P07A | 0 | 0.281945873 | 0.007739 |
| P08A | 0 | 0.508133817 | <1e-6 |
| P09A | 0 | 0.431528538 | 1.30E-05 |
| P10A | 0 | 0.168128839 | 0.062336 |
| P11A | 0 | 0.143276594 | 0.213631 |
| P12A | 0 | 0.043561404 | 0.67349 |
| P13A | 0 | 0.088531828 | 0.346438 |
| P01A | 1 | 0.419735569 | 0.003491 |
| P02A | 1 | 0.06814773 | 0.658923 |
| P04A | 1 | −0.124379351 | 0.493534 |
| P05A | 1 | 0.113506601 | 0.489726 |
| P06A | 1 | 0.429691697 | 0.001788 |
| P07A | 1 | 0.328041565 | 0.04651 |
| P08A | 1 | 0.142697377 | 0.326261 |
| P09A | 1 | 0.522358207 | 0.000105 |
| P10A | 1 | 0.125421761 | 0.389942 |
| P11A | 1 | 0.399401262 | 0.039162 |
| P12A | 1 | 0.183075563 | 0.32647 |
| P13A | 1 | 0.234323344 | 0.114713 |

**Appendix 1—table 4.** Table displaying individual pre (0) and post (1) challenge autocorrelation coefficient at 1 day lag for FEV1/FVC time series and associated p-values for the permutation tests conducted for assessment of the statistical significance of autocorrelation coefficients.

| Patient ID | Infection Status | 1 Day Lag FEV1/FVC Autocorrelation | p-value |
|---|---|---|---|
| P01H | 0 | 0.373461691 | 4.10E-05 |
| P03H | 0 | 0.532232344 | <1e-6 |
| P05H | 0 | 0.543561238 | <1e-6 |
| P06H | 0 | 0.699807531 | <1e-6 |
| P07H | 0 | 0.426087784 | 1.00E-06 |
| P08H | 0 | 0.656421903 | <1e-6 |
| P09H | 0 | 0.426194186 | <1e-6 |
| P11H | 0 | 0.875863699 | <1e-6 |
| P12H | 0 | 0.537767103 | <1e-6 |
| P13H | 0 | 0.122385415 | 0.240757 |
| P14H | 0 | 0.451348312 | 2.00E-06 |
| P15H | 0 | 0.065590774 | 0.469086 |
| P01H | 1 | 0.079120865 | 0.62836 |
| P03H | 1 | 0.082082101 | 0.591717 |
| P05H | 1 | −0.018492583 | 0.839892 |
| P06H | 1 | 0.307412139 | 0.028927 |
| P07H | 1 | −0.033268988 | 0.799478 |
| P08H | 1 | 0.156116543 | 0.144275 |
| P09H | 1 | 0.094516003 | 0.527785 |
| P11H | 1 | 0.215213314 | 0.13104 |
| P12H | 1 | 0.222717123 | 0.151388 |
| P13H | 1 | 0.221338946 | 0.183907 |
| P14H | 1 | −0.019789083 | 0.892886 |
| P15H | 1 | 0.0903505 | 0.516396 |
| P01A | 0 | 0.374048773 | 0.00036 |
| P02A | 0 | 0.287963001 | 0.003205 |
| P04A | 0 | 0.214051937 | 0.051278 |
| P05A | 0 | 0.579846615 | <1e-6 |
| P06A | 0 | 0.320989582 | 0.000793 |
| P07A | 0 | 0.497773172 | 2.00E-06 |
| P08A | 0 | 0.741401108 | <1e-6 |
| P09A | 0 | 0.61597128 | <1e-6 |
| P10A | 0 | 0.292115551 | 0.001014 |
| P11A | 0 | 0.510405399 | 5.00E-06 |
| P12A | 0 | 0.388760481 | 0.000102 |
| P13A | 0 | 0.428254592 | 1.00E-06 |
| P01A | 1 | 0.245018821 | 0.094571 |
| P02A | 1 | 0.339272642 | 0.024686 |
| P04A | 1 | 0.068042933 | 0.72154 |
| P05A | 1 | 0.123030146 | 0.430305 |
| P06A | 1 | 0.43121459 | 0.001864 |
| P07A | 1 | 0.209480725 | 0.195097 |

*Continued on next page*

*Appendix 1—table 4 continued*

| Patient ID | Infection Status | 1 Day Lag FEV1/FVC Autocorrelation | p-value |
|---|---|---|---|
| P08A | 1 | −0.213598501 | 0.119532 |
| P09A | 1 | 0.149361152 | 0.276512 |
| P10A | 1 | 0.077200085 | 0.598176 |
| P11A | 1 | 0.163280273 | 0.438443 |
| P12A | 1 | −0.019360275 | 0.916728 |
| P13A | 1 | 0.084470792 | 0.574956 |

**Appendix 1—table 5.** Table displaying individual pre (0) and post (1) challenge autocorrelation coefficient at 1 day lag for FVC time series and associated p-values for the permutation tests conducted for assessment of the statistical significance of autocorrelation coefficients.

| Patient ID | Infection Status | 1 Day Lag FVC Autocorrelation | p-value |
|---|---|---|---|
| P01H | 0 | 0.566972433 | <1e-6 |
| P03H | 0 | 0.436365475 | 1.10E-05 |
| P05H | 0 | 0.465971785 | 7.00E-06 |
| P06H | 0 | 0.640739895 | <1e-6 |
| P07H | 0 | 0.189205156 | 0.044844 |
| P08H | 0 | 0.90035042 | <1e-6 |
| P09H | 0 | 0.329621329 | 6.40E-05 |
| P11H | 0 | 0.758209436 | <1e-6 |
| P12H | 0 | 0.739080801 | <1e-6 |
| P13H | 0 | 0.042633778 | 0.679417 |
| P14H | 0 | 0.538541079 | <1e-6 |
| P15H | 0 | 0.275286764 | 0.003684 |
| P01H | 1 | 0.259356651 | 0.110413 |
| P03H | 1 | 0.179338037 | 0.241313 |
| P05H | 1 | 0.353179143 | 0.02498 |
| P06H | 1 | 0.169544692 | 0.236277 |
| P07H | 1 | 0.100365821 | 0.484466 |
| P08H | 1 | 0.231434951 | 0.112321 |
| P09H | 1 | 0.729801074 | <1e-6 |
| P11H | 1 | 0.185429238 | 0.220647 |
| P12H | 1 | 0.450384055 | 0.00285 |
| P13H | 1 | 0.020855678 | 0.901048 |
| P14H | 1 | 0.271008078 | 0.059202 |
| P15H | 1 | 0.326461968 | 0.028183 |
| P01A | 0 | 0.271994439 | 0.010138 |
| P02A | 0 | 0.438907563 | 5.00E-06 |
| P04A | 0 | 0.262286425 | 0.016387 |
| P05A | 0 | 0.712236062 | <1e-6 |
| P06A | 0 | 0.410534972 | 1.70E-05 |
| P07A | 0 | 0.203604433 | 0.057294 |
| P08A | 0 | 0.731352747 | <1e-6 |
| P09A | 0 | 0.590921928 | <1e-6 |

*Continued on next page*

*Appendix 1—table 5 continued*

| Patient ID | Infection Status | 1 Day Lag FVC Autocorrelation | p-value |
|---|---|---|---|
| P10A | 0 | 0.229274362 | 0.012014 |
| P11A | 0 | 0.299616931 | 0.009238 |
| P12A | 0 | 0.339086145 | 0.000807 |
| P13A | 0 | 0.439003821 | <1e-6 |
| P01A | 1 | 0.218598655 | 0.140146 |
| P02A | 1 | 0.270949027 | 0.076421 |
| P04A | 1 | −0.073928076 | 0.69089 |
| P05A | 1 | 0.293313312 | 0.062662 |
| P06A | 1 | 0.250391409 | 0.074608 |
| P07A | 1 | 0.189131405 | 0.254627 |
| P08A | 1 | 0.178045115 | 0.225613 |
| P09A | 1 | 0.445823622 | 0.001069 |
| P10A | 1 | −0.001582542 | 0.991459 |
| P11A | 1 | 0.456235007 | 0.025106 |
| P12A | 1 | 0.235389493 | 0.193009 |
| P13A | 1 | 0.123712298 | 0.408539 |

**Appendix 1—table 6.** Table displaying individual pre (0) and post (1) challenge autocorrelation coefficient at 1 day lag for PEF (% predicted) time series and associated p-values for the permutation tests conducted for assessment of the statistical significance of autocorrelation coefficients.

| Patient ID | Infection Status | 1 Day Lag PEF (% pred.) Autocorrelation | p-value |
|---|---|---|---|
| P01H | 0 | 0.20931285 | 0.025684 |
| P03H | 0 | 0.472289454 | <1e-6 |
| P05H | 0 | 0.431529163 | 2.30E-05 |
| P06H | 0 | 0.159205269 | 0.061348 |
| P07H | 0 | 0.353319402 | 9.00E-05 |
| P08H | 0 | 0.5547174 | <1e-6 |
| P09H | 0 | 0.556029042 | <1e-6 |
| P11H | 0 | 0.395824905 | 1.00E-05 |
| P12H | 0 | 0.352609112 | 0.000503 |
| P13H | 0 | 0.35119991 | 0.000595 |
| P14H | 0 | 0.402389428 | 1.70E-05 |
| P15H | 0 | 0.54985376 | <1e-6 |
| P01H | 1 | 0.488137429 | 0.001568 |
| P03H | 1 | 0.036320181 | 0.812622 |
| P05H | 1 | −0.058331926 | 0.719869 |
| P06H | 1 | 0.369715029 | 0.010815 |
| P07H | 1 | 0.008216799 | 0.954193 |
| P08H | 1 | 0.256946281 | 0.070942 |
| P09H | 1 | 0.145828174 | 0.335955 |
| P11H | 1 | 0.245096978 | 0.104066 |
| P12H | 1 | 0.159620218 | 0.302733 |

*Continued on next page*

*Appendix 1—table 6 continued*

| Patient ID | Infection Status | 1 Day Lag PEF (% pred.) Autocorrelation | p-value |
|---|---|---|---|
| P13H | 1 | 0.1689537 | 0.271384 |
| P14H | 1 | 0.382046159 | 0.007018 |
| P15H | 1 | 0.193711859 | 0.227004 |
| P01A | 0 | 0.771975809 | <1e-6 |
| P02A | 0 | 0.178970869 | 0.073432 |
| P04A | 0 | 0.244996288 | 0.025515 |
| P05A | 0 | 0.185148797 | 0.039857 |
| P06A | 0 | 0.576945862 | <1e-6 |
| P07A | 0 | 0.448316304 | 1.10E-05 |
| P08A | 0 | 0.401445077 | 3.90E-05 |
| P09A | 0 | 0.385012408 | 0.000101 |
| P10A | 0 | 0.363852864 | 4.60E-05 |
| P11A | 0 | 0.116865011 | 0.282604 |
| P12A | 0 | 0.259095938 | 0.010966 |
| P13A | 0 | 0.330342847 | 0.000308 |
| P01A | 1 | 0.662163321 | <1e-6 |
| P02A | 1 | 0.202406559 | 0.184951 |
| P04A | 1 | 0.152123058 | 0.419598 |
| P05A | 1 | 0.063184538 | 0.703673 |
| P06A | 1 | 0.527353781 | 3.40E-05 |
| P07A | 1 | 0.24076991 | 0.140734 |
| P08A | 1 | 0.278277054 | 0.061281 |
| P09A | 1 | 0.721164295 | <1e-6 |
| P10A | 1 | 0.282769697 | 0.04905 |
| P11A | 1 | 0.082807179 | 0.477035 |
| P12A | 1 | 0.199423959 | 0.28367 |
| P13A | 1 | 0.04844711 | 0.74762 |

**Appendix 1—table 7.** Table displaying individual pre (0) and post (1) challenge autocorrelation coefficient at 2 days lag for FeNo time series and associated p-values for the permutation tests conducted for assessment of the statistical significance of autocorrelation coefficients.

| Patient ID | Infection Status | 2 Days Lag FeNO Autocorrelation | p-value |
|---|---|---|---|
| P01H | 0 | 0.321709552 | 0.2069 |
| P03H | 0 | 0.472331673 | 0.0687 |
| P05H | 0 | 0.105726872 | 0.7053 |
| P06H | 0 | 0.046826106 | 0.7172 |
| P07H | 0 | 0.231741841 | 0.3424 |
| P08H | 0 | 0.204582651 | 0.4323 |
| P09H | 0 | −0.343752612 | 0.0976 |
| P11H | 0 | −0.074771481 | 0.6224 |
| P12H | 0 | −0.110383598 | 0.6075 |
| P13H | 0 | 0.209107713 | 0.4296 |
| P14H | 0 | 0.14699793 | 0.4745 |

*Continued on next page*

*Appendix 1—table 7 continued*

| Patient ID | Infection Status | 2 Days Lag FeNO Autocorrelation | p-value |
|---|---|---|---|
| P15H | 0 | −0.068181818 | 0.8054 |
| P01H | 1 | 0.314640112 | 0.3257 |
| P03H | 1 | 0.347121123 | 0.1962 |
| P05H | 1 | 0.124169987 | 0.6886 |
| P06H | 1 | 0.471311475 | 0.2051 |
| P07H | 1 | 0.21485574 | 0.4629 |
| P08H | 1 | −0.323361823 | 0.24 |
| P09H | 1 | −0.125 | 0.6982 |
| P11H | 1 | −0.289915966 | 0.332 |
| P12H | 1 | 0.352217742 | 0.2175 |
| P13H | 1 | −0.068493151 | 0.8668 |
| P14H | 1 | −0.091701293 | 0.7732 |
| P15H | 1 | 0.48125 | 0.0963 |
| P01A | 0 | 0.480394648 | 0.1001 |
| P02A | 0 | −0.244220459 | 0.2641 |
| P04A | 0 | −0.118700159 | 0.6458 |
| P05A | 0 | 0.103762095 | 0.6641 |
| P06A | 0 | 0.499676405 | 0.0264 |
| P07A | 0 | 0.735824043 | 0.0056 |
| P08A | 0 | −0.078914 | 0.7154 |
| P09A | 0 | −0.24540991 | 0.3058 |
| P10A | 0 | 0.004185217 | 0.9877 |
| P11A | 0 | 0.767363801 | 0.0018 |
| P12A | 0 | 0.46670556 | 0.0491 |
| P13A | 0 | 0.233558057 | 0.3441 |
| P01A | 1 | 0.579931923 | 0.0017 |
| P02A | 1 | 0.467260719 | 0.1566 |
| P04A | 1 | −0.291343669 | 0.3435 |
| P05A | 1 | −0.183087917 | 0.584 |
| P06A | 1 | 0.465145092 | 0.1357 |
| P07A | 1 | 0.272687609 | 0.3678 |
| P08A | 1 | 0.574961598 | 0.0605 |
| P09A | 1 | 0.360470496 | 0.2155 |
| P10A | 1 | 0.595921815 | 0.0443 |
| P11A | 1 | 0.685927733 | 0.0121 |
| P12A | 1 | 0.546008265 | 0.0609 |
| P13A | 1 | 0.632651662 | 0.0136 |

## Materials and Methods

### 1. Participant cohort

24 study subjects were recruited among which 12 were healthy volunteers and the other 12 steroids naïve (or stopped using steroids 6 weeks prior to the study) mild to moderately persistent asthmatics.

The inclusion and exclusion criteria for participants to enter the study were as follows:

## 1.1 Inclusion criteria

*Asthma patients* were selected using the following inclusion criteria:

- Age 18–50 years
- History of episodic chest tightness and wheezing
- Intermittent or mild to moderate persistent asthma according to the criteria by the Global

Initiative for Asthma (Global Initiative of Asthma. www.ginasthma.org)

- Non-smoking or stopped smoking more than 12 months ago and five pack years or less
- Clinically stable, no exacerbations within last six weeks prior to study
- Steroid-naïve or those participants who are currently not on corticosteroids and have not taken any corticosteroids by any dosing-routes within 6 weeks prior to the study or only using on-demand reliever therapy
- Baseline pre-bronchodilator $FEV_1$ $\geq$70% of predicted (*Quanjer et al., 1993*)
- Airway hyperresponsiveness, indicated by a positive acetyl-ß-methylcholine bromide (MeBr) challenge with $PC_{20}$ $\leq$9.8 mg/ml (*Sterk et al., 1993*)
- Positive skin prick test (SPT) to one or more of the 12 common aeroallergen extracts, defined as a wheal with an average diameter of $\geq$3 mm
- No other clinically significant abnormality on history and clinical examination
- Able to give written and dated informed consent prior to any study-specific procedures

*Healthy subjects* were selected using the following inclusion criteria

- Age 18–50 years
  - Non-smoking or stopped smoking more than 12 months ago and five pack years or less. Steroid-naïve, non-atopic participants who are currently not on any maintenance (subjects using oral contraceptives can be accepted)
- Baseline $FEV_1$ $\geq$80% of predicted
  - Negative acetyl-ß-methylcholine bromide (MeBr) challenge or $PC_{20}$ $\geq$19.6 mg/ml
  - Negative skin prick test (SPT) to one or more of the 12 common aeroallergen extracts
  - Negative history of pulmonary and any other relevant disease
  - Able to give written and dated informed consent prior to any study-specific procedure.

## 1.2 Exclusion criteria

Potential subjects who meet any of the following criteria were excluded from participation in the study:

- Women who are pregnant, lactating or have a positive urine pregnancy test at baseline visit
- Participation in any clinical investigational drug treatment protocol within the preceding five half-lives (or 12 weeks, if the half-life is unknown) before the screening visit of this study.
- Concomitant disease or condition which could interfere with the conduct of the study, or for which the treatment might interfere with the conduct of the study, or which would, in the opinion of the investigator, pose an unacceptable risk to the patient

Furthermore, the following additional exclusion criteria were used in phase 2 of the study:

- RV16 titer >1:8 in serum, measured at screening (visit 1) and also at the visit prior to the challenge
- History of clinically significant hypotensive episodes or symptoms of fainting, dizziness, or light-headedness
- History of an asthma exacerbation within the last 6 weeks prior to the study
- Has had any acute illness, including a common cold, within 4 weeks prior to visit 1
- Close contact with young children or with any immunosuppressed patients
- Has donated blood or has had a blood loss of more than 450 mL within 60 days prior to screening visit or plans to donate blood during the study
- Positive for any virus in nasal lavage at visit immediately prior to rhinovirus challenge.

## 1.3 Sample size calculation

The study is explorative in nature and is based on some first estimates of fluctuating inflammatory biomarkers. A definite sample size for all biomarkers measured was not possible to be accurately

determined. Conventionally, power calculations are based on comparison of groups using a single magnitude. This study was different in this regard, as multiple measurements were incorporated in a longitudinal setting to assess the temporal dynamics of different biomarkers before and after challenge with rhinovirus. Hence a proper a priori power estimation was not feasible due to not yet developed statistical tools.

However, A sample size of 12/12, based on previous studies with lesser data points, was calculated to identify biomarkers that have a strong, clinically relevant, sensitivity to the changes of disease stability over time (*Turner et al., 2006a*; *Turner et al., 2006b*). This sample size could probably miss capturing the changes in biomarkers with weak effect sizes. Nevertheless, this sample size has for all biomarkers screened provided evidence on long time variability, correlation and cross-correlation between biomarker properties, and susceptibility to strong viral external challenges. These three dimensions are a first estimate to understand the degree of variability and complexity of any disease process.

## 2. Study Design

The project represents a prospective observational, follow up study including patients with asthma and healthy controls with an experimental rhinovirus intervention in between.

### 2.1 Screening Visit

First, the subjects were asked to sign an informed consent form after which they were examined for inclusion and exclusion criteria along with medical history and scoring of adverse events and concomitant medication. Spirometry, Methacholine challenge and Skin Prick Test along with physical examinations were performed meticulously.

### 2.2 Run-in Phase

During this phase adverse events and medication were recorded along with physical examination. Exhaled nitric oxide (FeNO), nasal lavage, measurements were performed to acquaint the study subjects with the study procedures. Starting from the run-in, once-daily home-monitoring by Asthma Control Questionnaires, symptom scores and twice daily maneuvers by the pocket spirometer were executed.

### 2.3 Baseline visit

The medication history (also adverse events and concomitant medication) was carefully recorded and thereby routine FeNO, and nasal lavage were measured. Pregnancy test was also performed.

### 2.4 Study period

- Phase 1: The study participants were asked to visit the hospital clinic thrice weekly for the aforementioned measurements on Monday, Wednesday and Fridays, during the first 60 days.
- Phase 2: The participants were again followed up for the same measurements for the next 30 days with the same sampling frequency after being inoculated nasally with the common cold virus Rhinovirus 16, thereby mimicking the trigger for loss of asthma control or a mild exacerbation.

All participants in the study were screened for the presence of respiratory viruses just before the rhinovirus challenge, using Polymerase Chain Reaction confirmation on nasal lavage samples. This was performed in order to rule out a concomitant infection of the respiratory airways with viruses other than the one used in our study.

### 2.5 End of study visit

All measurements were repeated at the end of each phase.

The schematic representation of the phases of the study is mentioned below in *Appendix 1—figure 15*.

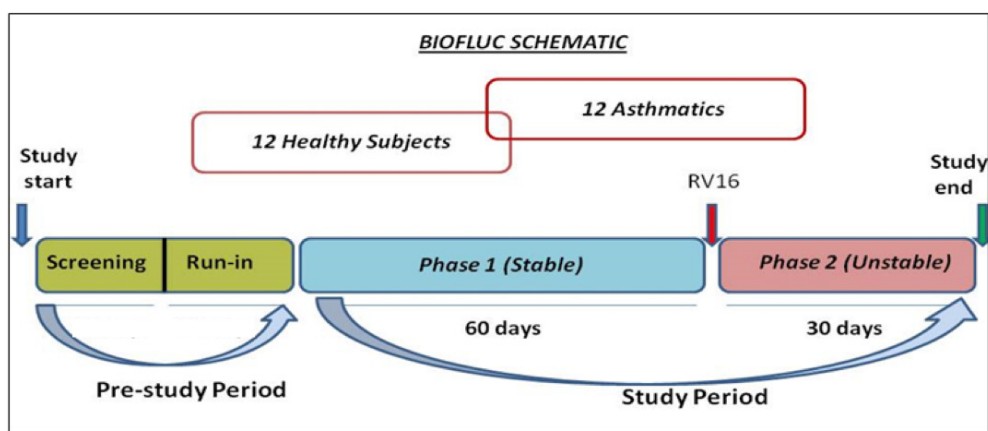

**Appendix 1—figure 15.** Schematic representation of the study design.

## 3. Measurement and collection of Biomarkers
### 3.1.1 Asthma Control Questionnaire

The Asthma Control Questionnaire (ACQ) was used in this study to assess the symptoms before and after Rhinovirus challenge along with lung function coupled to a hand-held spirometer device. Though widely used and well validated, it covers a 7 day time span and may not be optimal for a challenge protocol wherein symptoms change daily. Hence ACQ was used on a daily basis to record the daily changes in symptoms. Subjects complete the diary on rising in the morning and retiring at bedtime.

### 3.1.2 Skin prick test

Allergy skin prick tests were performed based on the position paper by the European Academy of Allergy and Clinical Immunology (EAACI) (*Dreborg and Frew, 1993*). The test was done by placing a drop of each of the 12 solutions containing common aeroallergens on the skin, followed by needle pricks by small SPT-lancets. Histamine was used as a positive control and saline solution as a negative control. The test result was considered positive if the skin develops a red, itchy area, called a 'wheal', with an average diameter of at least 3 mm, 15 min after the prick. The outline of all wheals was marked with a water-soluble marker and transferred to a test result from using adhesive tape, so as to facilitate measuring of the diameter of the wheal. All test result forms were archived. If the patient has been using anti-allergic medications, they were instructed to stop 3 days prior to the test in the clinic.

### 3.1.3 Spirometry

Spirometry was performed using a daily calibrated spirometer according to European Respiratory Society (ERS) recommendations. Spirometry was done only once on the screening visit at the clinic. There was no use of bronchodilators (pre/post measurements) in the study. The subjects were asked not to use reliever medications at least 6 hr before the test starts in the clinic. An experienced lung function analyst from the AMC performed the lung function tests throughout the study to reach optimal performance and to enhance reproducibility. Spirometry (FEV1) results were printed out and documented. Home monitoring of morning and evening lung function were done by Peak Expiratory Flow Meter (PEF, Micro Diary, CareFusion).

Waking and bedtime FEV1 separately are of interest due to the exaggerated diurnal airflow variation seen in asthmatics. Subjects were encouraged to measure FEV1 at consistent times (upon waking between approximately 6am-9am and prior to bedtime between approximately 8pm-11pm, respectively – always prior to any bronchodilator therapy).

Forced expiratory flow 25–75% (FEF 25–75) was measured. Patients were systematically instructed to perform the home monitoring PEF measurements. The electronic PEF devices were tested and compared to the lab spirometry measurements.

### Home monitoring

The morning and evening lung function measurements at home was done by Peak Expiratory Flow Meter (PEF, Micro Diary, CareFusion).

The following routine lung function indices were measured along with Asthma Control Questionnaire

FEV1: Forced Expiratory Volume in 1$^{st}$ second

FVC: Forced Vital Capacity

FEV1/FVC: Ratio of the amount of air exhaled in 1$^{st}$ second to the amount of air exhaled during a maximal expiration.

PEF: Peak Expiratory Flow

All the home recordings were monitored regularly (every 2 weeks) for uniformity and consistency of data. Stringent quality control of home lung function measurements was done at each visit.

### 3.1.4 Exhaled nitric oxide

Measurements of fractionated exhaled nitric oxide (FENO) were performed (one measurement per subject) using the NIOX MINO (Aerocrine AB, Sweden). This analyser has an accuracy of ±5 ppb or max 10%. The precision is <3 ppb of a measured value <30 ppb and ±10% of measured value >30 ppb. Every individual subject was measured using a single measurement as standardised in the hospital.

Subjects were asked to perform a slow vital capacity manoeuvre with a standardised expiratory flow of 50 ml/sec for as long as possible. Positive mouth pressure was applied to close the velum and prevent contamination with NO from nose and paranasal sinuses. Expired gas was sampled continuously at the mouth piece and mean FENO values at the end expiratory plateau were calculated.

### 3.1.5 Methacholine challenge

Methacholine is a cholinergic synthetic analogue of acetylcholine and acts directly on the airway smooth muscle resulting in bronchoconstriction. Measurement of airway responsiveness can be done using incremental inhaled doses of methacholine as bronchoprovocation test. This challenge test was performed using MeBr (acetyl-β-methylcholine bromide) according to the standardized tidal volume method that is operative in routine clinical diagnostics in the hospital (*Sterk et al., 1993*).

### 3.1.6 Nasal lavage

Nasal lavage was collected from the study participants once weekly before rhinovirus challenge and was up scaled to thrice weekly after the challenge at the clinic.

Eight ml of saline solution was introduced into the nasal cavity by a catheter and maintained for 10 min before recovery, followed by filtration and removal of mucous and cells by centrifugation. Standardized washings collected from the nose was used for biomarker analyses.

### 3.1.6.1 Cell and slide preparation

A cell suspension of 200,000 cells/ml were prepared in PBS. 2 sets of Cytospin apparatus (Thermo Shandon Cytospin 4) were assembled and the cytospin filter was pre wet by spinning at 550 rpm for 1 min with 50 µl PBS. 100 µl of the cell suspension was added to each slide and centrifuged at 450 rpm for 2 min. The quality of slides and the density of the cells were evaluated using a phase contrast microscopy. If the cell density was too high or low, more slides were prepared using an adjusted volume of the cell suspension. A target of 4 good slides with no overlapping or clumping of cells was set to improve the overall quality and consistency.

If the total cell count (for both slides together i.e. less than 7500 cells/slide) was less than or equal to 15000 cells, then the sample was centrifuged and re-suspended in 100 µl and was loaded onto a single block.

The prepared slides were stained as soon as possible with Diff Quick one stain (Medion Diagnostics), dried and mounted on Depex.

### 3.1.6.2 Nasal cell cytology

A differential cell count on a maximum of 400 inflammatory cells were performed. Numbers of eosinophils, neutrophils, macrophages/monocytes, lymphocytes, columnar and squamous epithelial cells were recorded. A report was prepared with the overall counts, the quality of cytospin and additional aspects such as the presence of mucous, cell debris, inclusions, eosinophil granules etc.

### 3.1.7 Blood venapunction

15 ml of venous blood was collected at each scheduled visit (once weekly) to determine whether circulating antibodies against RV16 are present, for hematology, and other immunological biomarker assays. Collection of blood was done in standard EDTA and non-EDTA tubes for specific purposes. 10 ml of it was immediately be followed by centrifugation to obtain plasma [2000 g for 10 min (min) at room temperature, RT] aiding in removal of RBCs and WBCs. The supernatants were aliquoted for biomarker hunt.

### 3.2 Rhinovirus challenge

In this study we exposed the volunteers to a mild dose (100 TCID 50) of RV16 using a validated approach which has been previously shown to be sufficient in inducing mild cold symptoms and decrease in lung function.

We used the GMP RV16 stock that has been tested previously by the medical ethical commission at the hospital in AUMC, and also by the U-BIOPRED showing efficacy in terms of cold symptoms and viral replication at 100 TCID50, which is part of the accompanying IMPD. This GMP RV16 stock has been prepared from a seed virus in extensively characterized human volunteers as described in METC 2010_310 and was expanded and aliquoted by Charles River Laboratories (USA) under GMP conditions. This preparation was considered safe for in vivo testing in human volunteers during a scientific advice meeting at BFarM (Bonn, Germany, April 30, 2013). This testing was carried out as per the approved protocol from the Amsterdam UMC medical ethical committee.

## 4. Statistical and computational Analysis

All computations were done using R (*R Development Core Team, 2018*) version 3.5.2 together with the following R-packages: car (*Fox and Weisberg, 2011*), reshape2 (*Wickham, 2007*), openxlsx (*Walker, 2018*), lubridate (*Grolemund and Wickham, 2011*), emdist (*Urbanek and Rubner, 2012*), gplots (*Warnes et al., 2014*), ape (*Paradis et al., 2004*), ggdendro (*de Vries and Ripley, 2013*), cluster (*Maechler et al., 2018*), factoextra (*Kassambara and Mundt, 2017*), philentropy (*Drost, 2018*), dendextend (*Galili, 2015*), and plyr (*Wickham, 2011*). Statistical tests resulting in a p-value less or equal to 0.05 were regarded as significant.

### 4.1 Assessment of differences: Pre- vs. post-viral-challenge

Consecutive measurements of a given biomarker prior to and after the viral challenge resulted in pre- and post-challenge time series for each cohort participant. Except for the calculation of transient changes, the time series were treated as empirical distributions, thus disregarding the chronological order of the measurements.

The empirical distribution of any biomarker before and after viral challenge was compared using Kolmogorov Smirnov test. A participant is considered a responder with respect to a given biomarker if the outcome of the Kolmogorov-Smirnov test results in a p-value<=0.05. Differences in the variance between the pre- and post-challenge distributions were assessed using Levene's test.

The empirical distributions of a given biomarker, prior to and after the viral challenge, were compared to each other using the Earth Mover's Distance (EMD), see *Figure 2* in the Main Manuscript. The resulting pair-wise distances between distributions were then used for hierarchical clustering of pre- and post-challenge distributions. Our clustering approach makes use of the entire collection (distribution) of values measured before and after the challenge, respectively, and does not amalgamate the information into a single magnitude (e.g., the mean value). This method unveils subtler differences and similarities between the participants' measurements that are less likely to be captured by conventional methods based on averages. Therefore, this part of our methodology is based

on the distributional properties of each participant's measurements and neglects the time dimension.

## 4.2 Clustering approach

In our approach, for each biomarker, the measurements collected before and after the viral challenge were used to construct the individual empirical distribution of measurements for each study participant, before and after the challenge, respectively.

We performed agglomerative hierarchical clustering (*Gan et al., 2007*) of the aforementioned empirical distributions of biomarkers. Within the hierarchical clustering algorithm, the distances between the distributions were calculated using the Earth Mover's Distance (*Rubner et al., 1998*), and the agglomeration procedure was done according to Ward's minimum variance method (*Gan et al., 2007*). Intuitively speaking, the Earth Mover's Distance contemplates the pair of distributions to be compared as piles of sand and measures the effort that it would take to shovel one distribution into the shape and position of the other (see below for a more detailed description of this method).

## 4.3 Calculation of short-term/transient changes

For each participant individually, throughout the entire period of observation, the relative change of each biomarker taking place within time intervals of 10 days was calculated. This was done throughout the entire period of observation considering all possible time intervals consisting of 10 consecutive days, see *Figure 3* in the Main Manuscript. The rationale for this is as follows. Any changes taking place within a period of 10 days that contained the day of the viral challenge need to be interpreted and understood in the context of any changes taking place within a period of about 10 days during the entire pre-challenge phase of the study.

For each participant, the day of the challenge (i.e., the day of the inoculation) was labeled as 'day 0'. All days between recruitment and the day of the challenge were labeled with negative integer numbers. All days between the day of the challenge and the participant's final visit were labeled with positive integer numbers. Given that measurements were not conducted every day on a given participant (see study design above), for each biomarker considered in this study and for each participant separately, interpolation was used in order to have, for every biomarker one value for every day. Except for the total cell count in nasal lavage fluid, and the percentages of eosinophils and neutrophils in nasal lavage fluid, which were linearly interpolated, all biomarker time series were interpolated using cubic splines with natural boundary conditions (see, e.g., *Bartels et al., 1995*). These interpolated values were only used for the assessment of the short-term/transient response.

The time interval of 10 days was decided based on prior knowledge where differences in signals from biomarkers (lung function etc.) usually take not less than a week to subside

As a sensitivity analysis, the same calculations were repeated using a 7 day time interval (data not shown). The outcomes were very similar to the ones obtained using the 10 day interval.

Consequently, a 10 day time period was considered optimal to calculate the short term/transient changes.

## 4.4 Characterization of the dendrogram clusters

The clusters obtained using the clustering dendrogram were tested for enrichment in or depletion of healthy or asthmatic participants, and/or for enrichment in or depletion of pre- or post-challenge distributions.

The relative location of leaves in the clustering dendrogram was quantitatively evaluated using the cophenetic distance (*Sokal and Rohlf, 1962*). The cophenetic distance between two leaves of a dendrogram is defined as the height of the dendrogram at which the two largest branches that individually contain the two leaves merge into a single branch.

For every cohort participant and any given biomarker there is a pre-challenge and a post-challenge time series, which we call the participant's pre- and post-pair. If the disruption caused by the viral challenge is not strong enough, the pre- and post-challenge distributions of a given participant will tend to cluster together. Therefore, a cluster in which pre- and post-pairs are closely located in terms of the cophenetic distance within the dendrogram, represents a subgroup of participants for

which the viral challenge caused a relatively weaker disruption, at least with respect to the biomarker under scrutiny.

Two dendrogram leaves are called neighbors if their mutual cophenetic distance is equal to the minimum of all cophenetic distances from one of the leaves to all the other leaves in the dendrogram. If this condition is fulfilled for both leaves simultaneously, then the two leaves form a two-element cluster in the dendrogram. If the condition is only fulfilled for one of the leaves, the two are still considered neighbors, even if this is not always visually obvious from inspecting the dendrogram (see Figure below).

Under the null-hypothesis that the branching in the dendrogram is the result of a purely random process, the number of neighboring pre- and post-pairs to be expected just by chance within a given cluster can be estimated by simply permuting the labels of the leaves in the dendrogram and counting the number of neighboring pre- and post-pairs. This permutation test is used for calculating the empirical p-values displayed in *Tables 1* and *3* in the Main Manuscript.

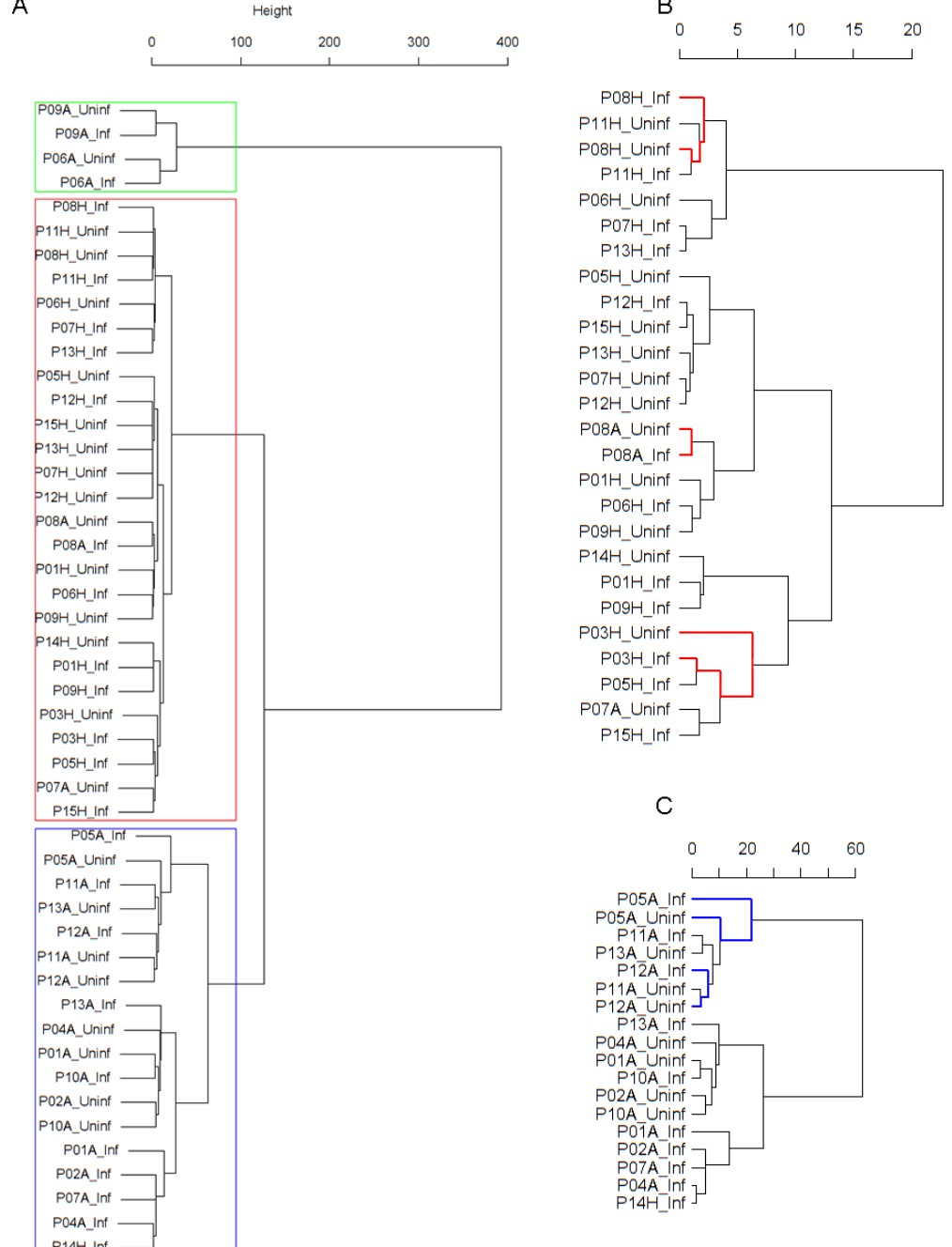

**Appendix 1—figure 16.** Panel A: Cluster dendrogram obtained via hierarchical clustering of the participants' pre- and post-challenge time series of FeNO. The distance between any two-time series was calculated using the EMD. Rectangles mark the clusters identified. Panel B displays a more detailed view of the second cluster. According to the definition of neighboring leaves provided in the text above, the leaves P03H_Uninf and P03H_Inf are neighbors in this dendrogram (highlighted in red). The reason for this is that the cophenetic distance from leaf P03H_Uninf to leaf P03H_Inf is equal to the minimum of all distances from leaf P03H_Uninf to all other leaves in the dendrogram. This is not the case for the leaf P03H_Inf. However, the fact that this condition holds for at least one of the two leaves renders them neighboring. Leaves P08H_Inf and P08H_Uninf, and P11H_Inf and P11H_Uninf, are neighbors, respectively (the latter pair is not marked in color). P08A_Inf and P08A_Uninf are also neighbors; In this case, the minimum condition is fulfilled by both leaves for of this leaf-pair. This is why the two leaves form a two-element cluster in the dendrogram. Panel C displays a more detailed view of the third cluster. Analogous information about neighboring leaves

in Cluster three is highlighted in blue. However, as opposed to Cluster 2 (depicted in Panel B), the amount of neighboring leaves in Cluster three is not statistically significant (permutation test, see *Table 2* in the Main Manuscript).

## 4.5 Soft Bootstrapping

As elucidated in *Mucha and Bartel (2014)*, when resampling with replacement from a given time series in order to generate a bootstrap replicate of the same time series, the relative frequencies of the values in the original time series were adjusted in the sense of soft bootstrapping as follows:

1. The relative frequencies of the values in the original time series were sorted in ascending order: $p_1 \leq p_2 \leq p_3 \leq \ldots \leq p_m$, where $m$ is the number of different values in the time series.
2. The smallest $(p_1)$ and the strictly second-smallest $(p_q)$ relative frequencies were adjusted using a softness parameter (see *Mucha and Bartel, 2014*) $\delta = 0.008$ according to the following formula:
$$p_i' = p_i + \delta \quad \text{for} \quad i \in \{1, \ldots, q\}$$
3. The remaining relative frequencies were adjusted according to the following formula:
$$p_i' = p_i - \frac{q\delta}{m-q} \text{ for } i > q$$

For time series with $m < 3$ the corresponding relative frequencies were left unchanged during the soft bootstrapping iterations.

## 4.6 The Earth Mover's Distance

The Earth Mover's Distance (*Rubner et al., 1998*) is a method for quantifying the dissimilarity between two probability distributions. Intuitively speaking, the EMD contemplates the pair of distributions to be compared as piles of sand and measures the *minimal* effort that it would take to shovel one distribution into the shape and position of the other.

In practice, two distributions will be given by two representative samples, which can be written as lists of pairs $\{(v_1, w_1), \ldots, (v_n, w_n)\}$ and $\{(c_1, f_1), \ldots, (c_n, f_m)\}$. Each pair $(v_i, w_i)$ corresponds to a value $v_i$ and its relative frequency $w_i$ in the sample. If we translate the above described intuitive approach into numbers, the problem becomes a well-known transportation problem (*Hitchcock, 1941*): Suppose that $n$ suppliers are located at positions $v_1, \ldots, v_n$, respectively, and each one has a given amount of goods $w_i$. Furthermore, they are required to supply $m$ consumers, located at positions $c_1, \ldots, c_m$, respectively, whereas each one has a given specific demand $f_i$. For each supplier-consumer pair $(v_i, w_i)$ and $(c_j, f_j)$, the cost of transporting a single unit of goods is determined by the distance $d$ $(v_i, c_j)$ between their locations. The transportation problem is then to find a least expensive pattern of flow of goods from suppliers to consumers that would satisfy the consumers' demand. Once the optimal pattern of the goods' flow has been found, the total cost is the corresponding EMD.

Mathematically, this transportation problem can be formalized as a linear programming problem, for which efficient solution algorithms were developed in the late 1940 s (see, e.g. *Hillier and Lieberman, 2010*).

## Discussion
### Utility of biomarker time series analysis

Do longitudinal measurements provide deeper insights into complex disease physiology as compared to single measurements or average values? In order to answer this question, we tried to reproduce the results obtained using each participant's entire collections of pre- and post-challenge biomarker measurements after collapsing them to the corresponding pre- and post-challenge individual average.

For example, in this cohort, FeNO time series have the ability to discriminate between healthy and asthmatics, and, within the group of asthmatics, between infected and uninfected (see *Figure 1* in the Main Manuscript). In order to investigate whether the average value would have a similar discriminative power, we calculated, for each participant, the average of their pre- and of their post-challenge series and used the absolute value of the difference between averages as a distance measure for clustering. The resulting dendrogram is depicted in Figure *Appendix 1—figure 17* below.

While the discriminative power between healthy and asthmatics is still given, the ability to distinguish between infected and uninfected within the group of asthmatics gets lost.

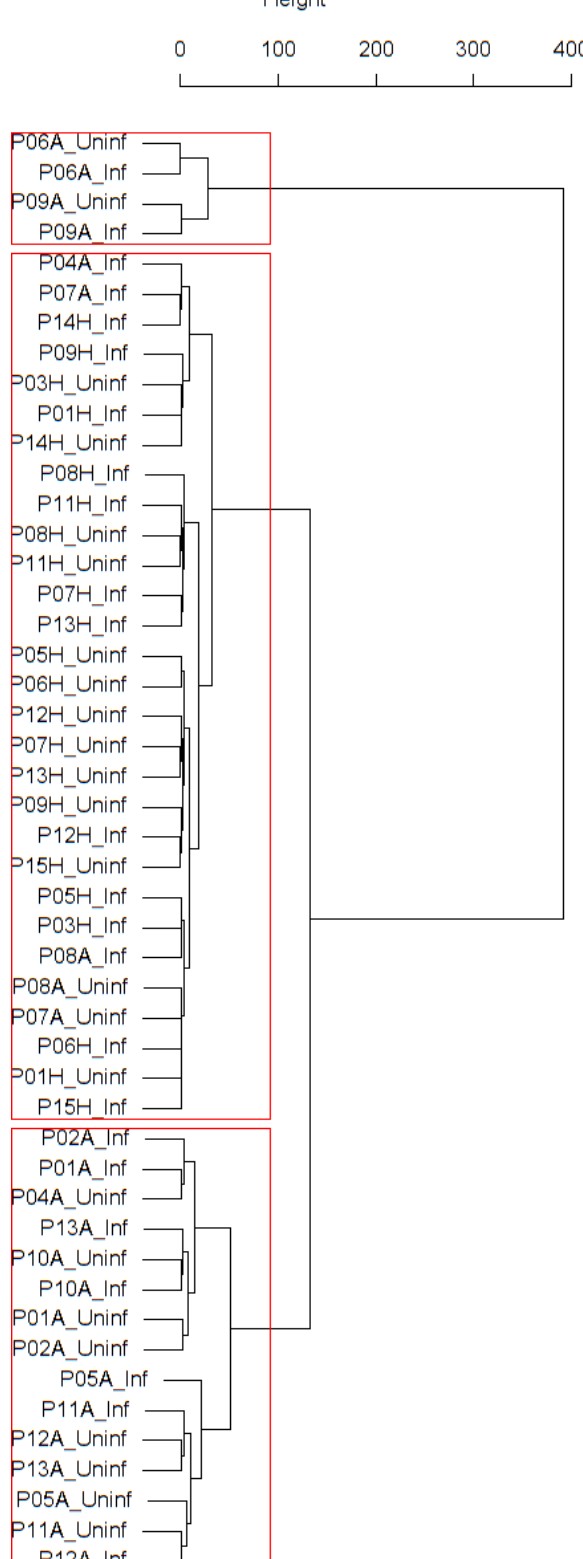

**Appendix 1—figure 17.** Dendrogram obtained from clustering the participants' pre- and post-challenge *average* value of FeNO using the EMD.

## Summary Statistics of biomarkers in the study

| | Min | 1st QU | Median | Mean | 3rd QU | Max |
|---|---|---|---|---|---|---|
| PEF (% predicted) | | | | | | |
| Healthy group before viral challenge | 63.31 | 84.63 | 91.52 | 92.16 | 100.97 | 119.01 |
| Healthy group after viral challenge | 63.42 | 78.42 | 91.92 | 90.44 | 102.39 | 118.84 |
| Asthmatics group before viral challenge | 57.62 | 82.45 | 88.64 | 86.77 | 94.14 | 105.09 |
| Asthmatics group after viral challenge | 47.67 | 78.49 | 88.44 | 85.01 | 95.21 | 109.98 |
| Normalized FEV1 | | | | | | |
| Healthy group before viral challenge | -2.7287 | -1.5404 | -0.9138 | -1.0405 | -0.7148 | 0.3588 |
| Healthy group after viral challenge | -3.0971 | -1.9662 | -1.3308 | -1.3310 | -0.8425 | 0.4305 |
| Asthmatics group before viral challenge | -2.4603 | -2.1253 | -1.0570 | -1.3044 | -0.8828 | -0.1823 |
| Asthmatics group after viral challenge | -3.0350 | -2.1484 | -1.1480 | -1.3369 | -0.8418 | 0.5686 |
| Normalized FVC | | | | | | |
| Healthy group before viral challenge | -2.19123 | -1.87550 | -1.66646 | -1.34813 | -0.83674 | -0.02098 |
| Healthy group after viral challenge | -3.0841 | -2.6730 | -2.1174 | -1.7966 | -0.9592 | 0.2229 |
| Asthmatics group before viral challenge | -3.0841 | -2.6730 | -2.1174 | -1.7966 | -0.9592 | 0.2229 |
| Asthmatics group after viral challenge | -3.726 | -1.840 | -1.319 | -1.483 | -0.668 | 0.194 |
| Normalized FEV1/FVC | | | | | | |
| Healthy group before viral challenge | -1.4796 | 0.1851 | 0.6670 | 0.6940 | 1.3305 | 2.1131 |
| Healthy group after viral challenge | -0.6591 | 0.1088 | 1.3446 | 1.1083 | 1.9541 | 2.5255 |
| Asthmatics group before viral challenge | -1.2123 | -0.5753 | -0.3160 | 0.1677 | 1.2150 | 1.7450 |
| Asthmatics group after viral challenge | -1.5441 | -0.4089 | 0.3237 | 0.4484 | 1.3195 | 2.5751 |
| FeNO | | | | | | |
| Healthy group before viral challenge | 9.435 | 12.779 | 13.872 | 14.608 | 15.594 | 22.227 |
| Healthy group after viral challenge | 7.545 | 10.807 | 16.143 | 15.841 | 19.350 | 25.909 |
| Asthmatics group before viral challenge | 16.57 | 35.67 | 46.21 | 61.97 | 55.68 | 181.30 |
| Asthmatics group after viral challenge | 18.36 | 29.80 | 42.59 | 61.42 | 58.73 | 181.18 |
| Cell density in $10^6$ per ml of nasal lavage fluid | | | | | | |
| Healthy group before viral challenge | 0.1975 | 0.4308 | 0.5569 | 0.9327 | 1.2431 | 2.4600 |
| Healthy group after viral challenge | 0.3436 | 0.6409 | 1.4536 | 1.6078 | 2.3982 | 3.7255 |
| Asthmatics group before viral challenge | 0.0375 | 0.2490 | 0.8489 | 1.4183 | 1.8562 | 5.0444 |
| Asthmatics group after viral challenge | 0.2873 | 0.9698 | 1.8445 | 2.7841 | 4.0850 | 10.8618 |
| Percentage of Eosinophils in nasal lavage fluid | | | | | | |
| Healthy group before viral challenge | 0.00000 | 0.00000 | 0.04375 | 0.95683 | 0.36979 | 9.03750 |
| Healthy group after viral challenge | 0.0000 | 0.0000 | 0.0500 | 1.2977 | 0.2909 | 14.1273 |
| Asthmatics group before viral challenge | 0.4091 | 7.2031 | 14.3950 | 16.9439 | 28.9778 | 35.9111 |
| Asthmatics group after viral challenge | 0.000 | 7.218 | 10.227 | 13.186 | 13.935 | 59.736 |
| Percentage of Neutrophils in nasal lavage fluid | | | | | | |
| Healthy group before viral challenge | 24.90 | 47.36 | 58.90 | 62.45 | 86.44 | 99.64 |
| Healthy group after viral challenge | 18.18 | 51.45 | 66.16 | 61.72 | 72.33 | 88.95 |
| Asthmatics group before viral challenge | 29.30 | 36.18 | 60.47 | 57.23 | 79.79 | 92.72 |
| Asthmatics group after viral challenge | 27.53 | 55.17 | 68.01 | 66.96 | 80.31 | 97.29 |

