## [Decision Letter]

**Acceptance summary:**

This study examines time courses of a number of phenotypic markers (both physiologic and biologic) related to asthma, following viral challenge in normal versus asthmatic subjects. Cluster analysis shows that the normals tended to retain the same characteristics after challenge that they exhibited prior to challenge, while the times series data from the asthmatics changed significantly after challenge. The authors interpret these findings as indicative of a general systems response to viral challenge that demonstrates a failure of restorative control in the asthmatics (termed homeokinesis).

**Decision letter after peer review:**

Thank you for submitting your article "Loss of adaptive capacity in asthmatics revealed by biomarker fluctuation dynamics on experimental rhinovirus challenge" for consideration by *eLife*. Your article has been reviewed by two peer reviewers, and the evaluation has been overseen by a Reviewing Editor and Neil Ferguson as the Senior Editor. The reviewers have opted to remain anonymous.

The reviewers have discussed the reviews with one another and the Reviewing Editor has drafted this decision to help you prepare a revised submission.

Essential revisions:

1) The main issue with the paper is that the reviewers and editors would like to become convinced that the major conclusions are of sufficient interest and in some way useful. Statements like "loss of adaptive capacity" and "failure of homeokinesis" may sound compelling in a general sense, they do not actually mean anything specific, and are thus unhelpful in elucidating the nature of asthma (see also comment 3).

2) Although the manuscript is of potential interest, in its current form it is still very difficult to follow. Data presentation, Introduction and Discussion should be thoroughly improved to make it more understandable to the non-expert reader. For example the Introduction should be rewritten. After the sentence 'We and others.…pertubation', first Homeokinesis should be explained (in fact the next paragraph until "We found evidence…"). Then focus on asthma. Please take care that the non-expert reader should be able to understand the text.

3) The study appears to have been skilfully performed in terms of data collection (indeed, the authors include world leaders in asthma research) and the application of statistical methods of cluster analysis. However, it is not clear what we have actually learned from this study. The time-series methods used are empirical in the sense that, while they can suggest that something is different between groups, little insight is provided about the mechanistic bases for these differences. For example, it is stated in the Discussion that "…these homeokinetic system characteristics of the lung likely contribute to the stability and dynamic phenotype of asthmatic disease." This seems to be a general statement one might make about any disease, but how it does it help us to understand asthma in particular. The fact that "FeNO and eosinophil time series were similar prior to and following the challenge in healthy subjects" is hardly a surprise given that asthma is typically characterized by type 2 allergic inflammation; there is clearly something about asthmatic subjects that makes them respond in this way when normal individuals do not, but saying that this is due to a failure of adaptive capacity is essentially just saying that there is something wrong with them. What we need to know is exactly what it is that is wrong with them, and concepts like loss of adaptive capacity and failure of homeokinesis do not seem to help much in this regard.

4) Perhaps some more conventional time-series analyses could be helpful. For example, it is stated that the difference found between the pre and post challenge time series were "not attributable to changes in the variance", which raises the question as what there were attributable to. There are two general possibilities – either the amplitude structures of the series (i.e., their histograms) were different, or their correlation structures (i.e., power spectra) were different (or perhaps both). Knowing more about this could help generate hypotheses about what was going on in the asthmatic subjects. For example, if the data in the asthmatic subjects were more strongly auto-correlated then maybe this would be indicative of some process related to a biorhythm. In any case, what is needed in this study is an analysis that leads to hypothesis about specific biologic mechanisms operative in asthma.

5) Discussion, second paragraph: "…provocation tests in asthma…implicitly rely on the assumption of loss of adaptive capacity…" We do not agree with this statement. Provocation tests in asthma rely on the fact that asthmatic subjects response more vigorously to airway challenge than do non-asthmatic subjects for a variety of reasons related to responsiveness of the airway smooth muscle, increased mucus secretions, and possibly airway wall remodeling. What does adaptive capacity have to do with this.

---

## [Author Response]

Essential revisions:1) The main issue with the paper is that the reviewers and editors would like to become convinced that the major conclusions are of sufficient interest and in some way useful. Statements like "loss of adaptive capacity" and "failure of homeokinesis" may sound compelling in a general sense, they do not actually mean anything specific, and are thus unhelpful in elucidating the nature of asthma (see also comment 3).

We thank the reviewers/editors for their comments and do understand the point. With this study we did not intend to achieve the ambitious goal of elucidating the nature of asthma by means of finding mechanistic explanations. Rather, we set out to investigate whether in asthmatics the adaptive capacity to a standardized environmental perturbation, such as an experimental viral challenge, is altered in comparison to healthy subjects. This has never been experimentally demonstrated using case and control groups, repeated biomarker sampling, and a carefully controlled challenge setting.

We realize that in the previous version of our manuscript we failed to provide precise definitions of homeokinesis and adaptive capacity, and how a loss of adaptive capacity can be quantitatively assessed. This is why the terms appeared somewhat unspecific and void. In the revised version of our manuscript we have now provided precise definitions, which will hopefully clarify the much humbler goal of this study.

The potential usefulness of this is that it may contribute to better phenotyping of individual patients based on disease dynamics. Snapshots of clinical and biological measures do not provide that. It is very likely that such phenotyping will have implications for asthma management. This obviously is a topic of a next prospective study. Apart from that, this study also helps us appreciate how the physiological system is regulated in asthmatic individuals as compared to healthy subjects. The regulation is estimated using adaptive capacity. This will help us better understand the dynamic nature and pathophysiology of respiratory diseases

2) Although the manuscript is of potential interest, in its current form it is still very difficult to follow. Data presentation, Introduction and Discussion should be thoroughly improved to make it more understandable to the non-expert reader. For example the Introduction should be rewritten. After the sentence 'We and others.…pertubation', first Homeokinesis should be explained (in fact the next paragraph until "We found evidence…"). Then focus on asthma. Please take care that the non-expert reader should be able to understand the text.

We thank the reviewers/editors for their feedback. We have now entirely rewritten the Introduction and parts of the Discussion, providing the definitions and explanations needed for non-expert readers to be able to understand the text. Furthermore, we have edited and extended the text and tables in the Results section in a way that hopefully will allow the readers to more easily grasp our main findings.

3) The study appears to have been skilfully performed in terms of data collection (indeed, the authors include world leaders in asthma research) and the application of statistical methods of cluster analysis. However, it is not clear what we have actually learned from this study. The time-series methods used are empirical in the sense that, while they can suggest that something is different between groups, little insight is provided about the mechanistic bases for these differences. For example, it is stated in the Discussion that "…these homeokinetic system characteristics of the lung likely contribute to the stability and dynamic phenotype of asthmatic disease." This seems to be a general statement one might make about any disease, but how it does it help us to understand asthma in particular. The fact that "FeNO and eosinophil time series were similar prior to and following the challenge in healthy subjects" is hardly a surprise given that asthma is typically characterized by type 2 allergic inflammation; there is clearly something about asthmatic subjects that makes them respond in this way when normal individuals do not, but saying that this is due to a failure of adaptive capacity is essentially just saying that there is something wrong with them. What we need to know is exactly what it is that is wrong with them, and concepts like loss of adaptive capacity and failure of homeokinesis do not seem to help much in this regard.

We thank the reviewers/editors for their comments. First, we believe that in the previous version of our manuscript we failed to provide precise definitions of homeokinesis and adaptive capacity, and how a loss of adaptive capacity can be quantitatively assessed. The latter is, methodologically speaking, not a trivial question given the fundamentally dynamic nature of physiologic systems. In the revised version of our manuscript we have now provided precise definitions, which will hopefully clarify the much humbler goal of this study.

Second, indeed the question raises what is driving these differences in dynamic behavior. Initially, we hesitated to speculate on this, but in the revised version we have added a sub-section entitled “Potential physiological and inflammatory mechanisms responsible for the biomarker dynamics observed in the group of asthmatics and the resulting reduction in adaptive capacity” to the Discussion to provide some potential mechanistic explanations that could be responsible for such differential dynamics amongst patients with asthma. We have substantiated our claims with references to published literature.

4) Perhaps some more conventional time-series analyses could be helpful. For example, it is stated that the difference found between the pre and post challenge time series were "not attributable to changes in the variance", which raises the question as what there were attributable to. There are two general possibilities – either the amplitude structures of the series (i.e., their histograms) were different, or their correlation structures (i.e., power spectra) were different (or perhaps both). Knowing more about this could help generate hypotheses about what was going on in the asthmatic subjects. For example, if the data in the asthmatic subjects were more strongly auto-correlated then maybe this would be indicative of some process related to a biorhythm. In any case, what is needed in this study is an analysis that leads to hypothesis about specific biologic mechanisms operative in asthma.

We thank the reviewers/editors for their comments. By definition, an experimental assessment of a physiological system’s level of adaptive capacity needs to be interventional, i.e., using an external perturbation. Consequently, the time series that we measured within this study are inevitably non-stationary after the viral challenge. On the other hand, most of the conventional time-series analysis methods require that the time series being analyzed is stationary (see, e.g., Manuca R. et al., Stationarity and nonstationarity in time series analysis, Physica D: Nonlinear Phenomena, Volume 99, Issues 2–3, 1996, Pages 134-161, ISSN 0167-2789). These circumstances limit the range of methodological approaches that can be applied. However, as the reviewer correctly points out, the histograms of the time series, that is, the empirical distribution of the data in a given time series is a meaningful quantitative way of looking at changes in the overall dynamics of the system being observed. This is precisely what we do when we use the Earth Mover’s Distance, which calculates the dissimilarity between two given time series in terms of the differences between their corresponding empirical distributions.

Moreover, we would like to thank the reviewer for their suggestion of looking at the autocorrelation properties of the time series. We carried out this analysis and indeed found a statistically significant difference between the healthy group and the asthmatics group in terms of the autocorrelation properties of FeNO time series after the challenge. Furthermore, for three lung function parameters there is a moderate positive autocorrelation before the challenge, which then disappears after the challenge. These findings are now described in a whole new sub-section in the Results section entitled “Autocorrelation properties of the biomarker time series”.

Finally, we try to discuss possible biological mechanisms operative in asthma in a new subsection of the Discussion entitled “Physiological interpretation of the group differences in autocorrelation properties of the biomarker time series”.

5) Discussion, second paragraph: "…provocation tests in asthma…implicitly rely on the assumption of loss of adaptive capacity…" We do not agree with this statement. Provocation tests in asthma rely on the fact that asthmatic subjects response more vigorously to airway challenge than do non-asthmatic subjects for a variety of reasons related to responsiveness of the airway smooth muscle, increased mucus secretions, and possibly airway wall remodeling. What does adaptive capacity have to do with this.

This is an interesting point. Even though we could argue that changes in responsiveness of the airway smooth muscle are a representation of a loss of adaptive capacity, we are taking the message that we need to be more specific here. In order to avoid confusion, we have removed this statement. Instead, as indicated above, we have added a more general discussion on the potential mechanistic background of our findings.